# Dual-Target Insight into Drug Discovery from Natural Products as Modulators of GLP-1 and the TXNIP–Thioredoxin Antioxidant System in Metabolic Syndrome

**DOI:** 10.3390/antiox14111364

**Published:** 2025-11-17

**Authors:** Peter Chinedu Agu, Appolonia Fulgence Yudas, Jun Lu

**Affiliations:** 1Key Laboratory of Luminescence Analysis and Molecular Sensing, Ministry of Education, College of Pharmaceutical Sciences and Chinese Medicine, Southwest University, Chongqing 400715, China; pc.agu@evangeluniversity.edu.ng (P.C.A.); appoloniafulgence@gmail.com (A.F.Y.); 2Department of Biochemistry, College of Science, Evangel University Akaeze, Abakaliki P.M.B. 129, Ebonyi State, Nigeria

**Keywords:** GLP-1, TXNIP, thioredoxin, natural products, antioxidant, metabolic syndrome, oxidative stress, multi-target therapeutics, drug discovery, computational strategies

## Abstract

Metabolic Syndrome (MetS), a cluster of interconnected metabolic abnormalities, poses a growing global health burden. A well-established therapeutic target for the diseases is the incretin hormone glucagon-like peptide-1 (GLP-1); however, synthetic agonists have drawbacks such as expense, injectable administration, and side effects. Concurrently, one of the main pathogenic characteristics of MetS is oxidative stress, in which the Thioredoxin-Interacting Protein (TXNIP)/thioredoxin system is a critical player. The strong evidence that natural compounds derived from plant, marine, and microbiological sources can simultaneously target the TXNIP–thioredoxin antioxidant axis and GLP-1 signaling is examined in this study. These substances can limit TXNIP expression and increase thioredoxin activity while also stimulating GLP-1 secretion, inhibiting dipeptidyl peptidase-4 (DPP-4), or acting as GLP-1 receptor agonists. A cycle of reinforcement is created by these two actions: Pancreatic β-cell activity and incretin responsiveness are improved by GLP-1-mediated TXNIP downregulation, which also strengthens antioxidant defense. However, translational development must overcome major pharmacological obstacles, especially those related to bioavailability, metabolic stability, and standardization, despite encouraging preclinical effectiveness. To speed up this translational process, integrative computational techniques (such as molecular docking, network pharmacology, and artificial intelligence) are strong tools for lead optimization and creation of hypothesis. Thus, natural products can provide a special chance to discover multi-target treatments that comprehensively address the oxidative and hormonal causes of MetS.

## 1. Introduction

Central obesity, insulin resistance, dyslipidemia, hypertension, and hyperglycemia are among the interrelated metabolic abnormalities that make up Metabolic Syndrome (MetS), which raises the risk of type 2 diabetes, cardiovascular disease, and metabolic-associated fatty liver disease [1]. The increasing incidence of MetS worldwide presents a serious public health concern, driven by sedentary lifestyles, inadequate nutrition, and continued urbanization [2]. Because traditional pharmacological management often requires combination medications that target different aspects of MetS, concerns regarding polypharmacy, excessive costs, and patient non-adherence are raised to underscore the need for comprehensive, multi-targeted treatment approaches [3].

Glucagon-like peptide-1 (GLP-1), an incretin hormone, has become a key treatment target for MetS. In response to dietary intake, intestinal L-cells secrete GLP-1, which increases the production of glucose-dependent insulin, inhibits the release of glucagon, postpones the emptying of the stomach, and increases satiety by activating the widely expressed GLP-1 receptor (GLP-1R) [4,5]. A multimodal strategy for enhancing glycemic control, promoting weight reduction, and controlling lipid metabolism is offered by this pleiotropic effect [6]. However, injectable delivery, gastrointestinal side effects, high cost, and difficulties with long-term adherence limit the use of synthetic GLP-1 receptor agonists [7,8].

Oxidative stress is identified as a key pathophysiological characteristic of MetS. It directly causes vascular damage, insulin resistance, and β-cell dysfunction [3,9]. The thioredoxin system, a key antioxidant defense mechanism that preserves cellular redox equilibrium, is a crucial regulator of oxidative stress. A key metabolic node, thioredoxin-interacting protein (TXNIP), functions as a negative regulator of thioredoxin and is a crucial mediator between oxidative stress, nutritional excess, and the activation of inflammatory pathways such as the NLRP3 inflammasome [10,11,12]. Thus, the TXNIP/thioredoxin axis is a promising therapeutic target for reducing oxidative damage in MetS since its overexpression is linked to poor glucose metabolism and increased apoptosis in pancreatic β-cells [13,14].

Critically, emerging reports reveal a robust mechanistic interconnection between these two systems by either direct or inferential evidence. For example, GLP-1 receptor activation has been shown to downregulate TXNIP expression via cAMP/PKA and PI3K/Akt signaling, thereby enhancing thioredoxin activity and attenuating oxidative stress and inflammation [12,15]. This crosstalk suggests that simultaneous targeting of both the incretin and redox systems could yield synergistic therapeutic benefits.

Given their extensive chemical diversity, traditional medical history, and innate polypharmacology, natural products offer a viable source for these multi-target medicines [16,17,18,19]. These plant, marine, and microorganism-derived compounds have the potential to simultaneously alter GLP-1 signaling and boost antioxidant defences through the TXNIP–thioredoxin pathway [18,20].

Therefore, this review aims to provide a comprehensive mechanistic insight into natural products as dual modulators of GLP-1 activity and the TXNIP–thioredoxin antioxidant system within the framework of MetS. It categorizes bioactive compounds according to their sources and mechanisms of action. It also critically evaluates the preclinical evidence supporting their efficacy and discusses the significant pharmacological challenges and translational opportunities in developing these multi-target therapies for MetS-related diseases. Therefore, the main goal is to elucidate the potential of a systems pharmacology approach that addresses both the hormonal and oxidative cores of MetS pathophysiology.

## 2. Overview of GLP-1 Modulation and TXNIP–Thioredoxin Antioxidant System

The pathogenesis of MetS is based on a complex interaction between persistent cellular stress and hormonal imbalance. According to this viewpoint, the body’s natural antioxidant defence mechanisms and the incretin hormone system are two of the most important and interrelated routes. Therefore, a thorough knowledge of these systems and their distinct functions in MetS will support the case for investigating them as dual targets with natural products.

### 2.1. The GLP-1 Pathway: A Multifunctional Regulator in MetS

#### 2.1.1. Physiology and Mechanism of Action

GLP-1 is an incretin hormone secreted by enteroendocrine L-cells in the distal ileum and colon in response to nutrient ingestion. As mentioned by Idoko et al. [20], its pleiotropic effects are mediated through binding to the widely expressed GLP-1 receptor (GLP-1R), a G-protein-coupled receptor. According to Sun et al. [21], upon activation, GLP-1 initiates a cascade of events crucial for metabolic homeostasis: (1) *Glucose-Dependent Insulin Secretion*: GLP-1 stimulates pancreatic β-cells to produce and release more insulin. Importantly, compared to many other anti-diabetic medications, this action is glucose-dependent, which means that it only increases insulin secretion during hyperglycemia. This results in a reduced risk of hypoglycemia. (2) *Suppression of Glucagon Secretion*: It inhibits the release of glucagon from pancreatic α-cells, reducing hepatic glucose production postprandially. (3) *Retarded Gastric Emptying*: GLP-1 delays the stomach’s emptying to cause a reduction in the rate at which foods enter the bloodstream. This leads to a smoother and lower postprandial glucose exit. (4) *Satiety Induction*: GLP-1 reduces food intake and facilitates weight reduction by acting on receptors in the brainstem and hypothalamus. This helps to create sensations of fullness and satiety. (5) *Preservation of β-Cell Function*: Research indicates that GLP-1 inhibits apoptosis and stimulates β-cell neogenesis and proliferation, which helps maintain functional β-cell mass over time [22,23].

#### 2.1.2. GLP-1 Dysregulation in MetS

People with type 2 diabetes and MetS frequently have reduced GLP-1 production in response to meals; this condition is referred to as “incretin deficiency.” Additionally, the short half-life (less than two minutes) of natural GLP-1 limits its bioactivity since the ubiquitous enzyme dipeptidyl peptidase-4 (DPP-4) quickly breaks it down. MetS is characterized by postprandial hyperglycemia and hyperglucagonemia, which are exacerbated by this impaired GLP-1 signaling [20,21].

#### 2.1.3. Current Synthetic Agonists and Their Limitations

According to Brunton [23], treatment for MetS and type 2 diabetes has been transformed by synthetic GLP-1 receptor agonists (like liraglutide and semaglutide) and DPP-4 inhibitors (like sitagliptin). Despite their great effectiveness, they have several significant drawbacks, including the administration type, gastrointestinal side effects, cost, and unknown long-term effects. Most first-generation agonists require daily or weekly subcutaneous injection, reducing patient acceptability. Because of the strong impact on stomach emptying, nausea, vomiting, and diarrhea are frequent, particularly during dosage escalation. These biologic therapies are significantly more expensive than traditional oral medications, limiting access. The long-term consequences of sustained, potent GLP-1R activation are still under investigation.

These limitations have fueled the search for natural compounds that can modulate the GLP-1 pathway in a more subtle, multi-targeted, and potentially safer manner [20].

### 2.2. The Antioxidant Defense System: Combating Oxidative Stress in MetS

#### 2.2.1. Oxidative Stress as a Pathogenic Core of MetS

An imbalance between the generation of Reactive Oxygen Species (ROS) and the body’s capacity to detoxify them leads to oxidative stress. Excessive dietary intake, hyperglycemia, and dyslipidemia are the main causes of MetS, a chronic low-grade inflammatory and oxidative stress condition [3]. Electrons escape, and a large amount of ROS is created when glucose and free fatty acids overburden mitochondrial oxidative phosphorylation. As discussed by González et al. [9], pathogenesis is directly aided by this oxidative environment by: (1) *Inducing insulin resistance*: ROS activate a cascade of stress kinases (e.g., JNK, IKKβ, PKC) that serine-phosphorylate Insulin Receptor Substrate (IRS) proteins, disrupting normal insulin signaling. (2) *Impairing β-cell function*: Pancreatic β-cells are exceptionally vulnerable to oxidative damage due to their low expression of antioxidant enzymes. ROS can lead to β-cell dysfunction and apoptosis. (3) *Promoting vascular dysfunction*: ROS oxidize LDL cholesterol, promote endothelial dysfunction, and stimulate vascular inflammation, accelerating atherosclerosis.

#### 2.2.2. The Thioredoxin System: A Key Guardian of Redox Balance

The thioredoxin system is one of the two major antioxidant systems in the cell (the other being the glutathione system). As per Lu and Holmgren [24], it consists of thioredoxin (Trx), thioredoxin reductase, and NADPH. Trx is a tiny, widely distributed protein having a disulfide/dithiol site that is redox active. As a strong protein disulphide oxidoreductase in its reduced form, Trx efficiently controls the activity of important signaling proteins by scavenging ROS directly and decreasing oxidized cysteine groups on those proteins. Maintaining a decreased cellular environment requires it. TrxR is an NADPH-dependent selenoenzyme that reduces oxidized Trx, thereby recycling it back to its active form. TXNIP is an internal negative Trx regulator. TXNIP binds to Trx’s reduced active site, blocking its redox-regulatory and antioxidant properties. TXNIP expression is highly sensitive to oxidative and metabolic stress and is upregulated by elevated levels of glucose, free fatty acids, and ROS itself.

#### 2.2.3. TXNIP as a Central Metabolic Node in MetS

Insightful from Alhawiti et al. [11] and Dagdeviren et al. [12], respectively, TXNIP is no longer viewed as a simple inhibitor but as a critical hub linking nutrient excess to metabolic dysfunction. It may exacerbate MetS pathology through multiple interconnected mechanisms. It can amplify oxidative stress by inhibiting Trx, leading to uncontrolled accumulation of ROS. Additionally, TXNIP is a key activator of the NLRP3 inflammasome, a complex that triggers the maturation and release of pro-inflammatory cytokines like IL-1β, which directly promote insulin resistance and β-cell damage. Moreover, high TXNIP expression in the pancreas is associated with increased β-cell apoptosis and reduced insulin secretion. In the liver, it influences gluconeogenic pathways.

Therefore, inhibition of TXNIP or enhancement of the entire thioredoxin system represents a powerful strategy to break the cycle of oxidative stress, inflammation, and metabolic dysfunction in MetS.

### 2.3. Interconnection Between GLP-1 and Antioxidant Pathways

Emerging evidence points to a strong mechanistic link between GLP-1 signaling and antioxidant defense, particularly through the TXNIP/Trx system. As per He et al. [25], GLP-1 receptor activation has been shown to diminish TXNIP expression in cardiomyocytes, hepatocytes, and pancreatic β-cells. This is primarily possible via the cAMP/PKA and PI3K/Akt signaling pathways, which suppress TXNIP gene transcription [26,27]. By reducing TXNIP, GLP-1 can enhance Trx activity, leading to a more robust antioxidant response characterized by lower ROS levels and attenuated activation of inflammatory pathways such as the NLRP3 inflammasome [28,29]. These antioxidant and anti-apoptotic effects represent a key cytoprotective mechanism, likely contributing to the preservation of β-cell function and the glycemic control-independent cardiovascular benefits associated with GLP-1 agonists [28].

Given their innate polypharmacology, natural compounds are uniquely positioned to achieve dual targeting of both the thioredoxin system and GLP-1 signaling [15]. As summarized in Table 1, such a dual-action approach would represent a highly synergistic and multifaceted strategy for addressing the root causes of MetS.

## 3. Categories of Natural Products as GLP-1 Modulators with Potential for TXNIP–Thioredoxin System Interconnection

The categories of natural product sources of agents that target GPL-1, as evidenced in the literature, are highlighted in Figure 1.

### 3.1. Plant-Derived Compounds

Pharmacologically active substances have long been found in plants, and many of these substances have promising effects on GLP-1 signaling pathways. In in vivo and in vitro investigations, phytochemicals such as flavonoids, terpenoids, alkaloids, and polyphenols have shown the ability to either activate GLP-1 receptors, block dipeptidyl peptidase-4 (DPP-4), or promote GLP-1 secretion. Furthermore, these substances also reduce oxidative stress, simultaneously targeting a key underlying mechanism of MetS.

For instance, quercetin, a flavonoid found in onions and apples, has been shown to increase GLP-1 secretion through calcium signaling in enteroendocrine L-cells [18,19,33]. Notably, quercetin also functions as a potent antioxidant that has the potential to downregulate TXNIP expression, thereby alleviating its inhibition of thioredoxin and enhancing cellular antioxidant capacity [34]. Similarly, genistein, an isoflavone derived from soybeans, enhances GLP-1 release and improves glucose tolerance in rodent models [35,36]. Genistein’s pharmacological effects are also linked to the suppression of oxidative stress pathways, including modulation of the thioredoxin system.

Berberine, an isoquinoline alkaloid from *Berberis* species, inhibits DPP-4 activity, thereby increasing endogenous GLP-1 levels [37]. The strong inhibition of TXNIP and consequent stimulation of the thioredoxin pathway, which lowers inflammation and death in pancreatic β-cells, may be a crucial complementary mechanism of berberine [38]. Extracts from *Momordica charantia* (bitter melon), *Camellia sinensis* (green tea; particularly epigallocatechin gallate, EGCG), and *Trigonella foenum-graecum* (fenugreek) have also been documented to exert GLP-1-enhancing properties [39,40]. These extracts reportedly act by either directly stimulating L-cells or by modulating gut microbiota, which indirectly influence incretin secretion [13,30]. Crucially, their rich polyphenolic content contributes significant antioxidant effects, with studies showing EGCG and fenugreek components can suppress TXNIP overexpression and bolster thioredoxin activity under conditions of metabolic stress [41,42].

Therefore, these plant-derived agents are attractive due to their availability, low toxicity, and multi-targeted action. Their unique capacity to simultaneously enhance GLP-1 signaling and potential to reinforce the TXNIP–thioredoxin antioxidant defense system makes them strong, holistic candidates for further development in the management of MetS.

### 3.2. Marine Natural Products Targeting GLP-1

A varied and mainly unexplored supply of new bioactive substances with possible metabolic advantages can be found in marine environments. Numerous of these bioactive substances have shown a special ability to interact with redox-sensitive targets such as the TXNIP–thioredoxin system as well as incretin pathways. Similarly, incretin-modulating characteristics have been demonstrated in the literature for natural compounds obtained from marine sources, including lipids, polysaccharides, and peptides. This is often paired with potent antioxidant properties.

For example, fucoxanthin, a carotenoid derived from brown seaweeds such as *Undaria pinnatifida*, has been reported to enhance insulin sensitivity and stimulate GLP-1 secretion in animal models [17,43,44]. Fucoxanthin may have mechanisms beyond its incretin effects, such as downregulating TXNIP expression and increasing thioredoxin activity, which would reduce inflammation and oxidative stress in adipose and hepatic tissues [45]. Sulfated polysaccharides (e.g., fucoidans, ulvans) extracted from red algae (*Gracilaria* spp.) and green algae (*Ulva lactuca*) exhibit DPP-4 inhibitory activity, thereby prolonging GLP-1 half-life [46,47,48,49]. These same polysaccharides also demonstrate potent antioxidant effects. These antioxidant effects imply the ability to scavenge ROS and modulate the expression of TXNIP, supporting the restoration of thioredoxin-mediated redox balance [50,51].

Furthermore, it is known that cyanobacteria and marine sponges generate brominated metabolites and cyclic peptides that interact with metabolic pathways related to GLP-1 biology; however, little is known about these substances [52,53]. However, early mechanistic studies suggest that several of these compounds, such as bromotyrosine derivatives, not only influence GLP-1 secretion but can also suppress TXNIP-mediated NLRP3 inflammasome activation [54]. These stress a dual mode of action targeting both metabolic and inflammatory arms of MetS.

Marine compounds’ distinct structural motifs, like halogenation, sulfation, and novel cyclic backbones, consistently provide useful scaffolds for semi-synthetic modification and the creation of innovative therapeutics that can simultaneously modulate GLP-1 signaling and strengthen the TXNIP–thioredoxin antioxidant axis. Marine natural materials are therefore positioned as prospective leads for multi-mechanistic therapies in MetS due to their dual targeting potential.

### 3.3. Microbial Metabolites Targeting GLP-1

In the search for multi-target drugs for MetS, microbial sources like probiotic bacteria, soil actinomycetes, and endophytic fungi are being increasingly recognized not only for their ability to modulate GLP-1 but also for their emerging role in regulating oxidative stress through the TXNIP–thioredoxin system.

Existing pieces of literature have shown that the gut microbiota ferment dietary fibers to create short-chain fatty acids (SCFAs) such as acetate, propionate, and butyrate [55,56,57]. These SCFAs are known to activate G-protein coupled receptors (GPR41 and GPR43) on L-cells, which in turn promote GLP-1 production [16,58]. Importantly, SCFAs, particularly butyrate, have been demonstrated to significantly downregulate TXNIP expression in intestinal cells, hepatocytes, and pancreatic β-cells. Inferentially, this suppression relieves the inhibition of thioredoxin, thereby enhancing cellular antioxidant defenses and reducing NLRP3 inflammasome activation, a key link between oxidative stress and metabolic inflammation [59,60].

Certain strains of *Lactobacillus* and *Bifidobacterium* have proved the ability to enhance GLP-1 secretion, modulate gut hormone release, and improve insulin sensitivity in host organisms [61]. The beneficial metabolic effects of these probiotics are increasingly attributed not only to incretin modulation but also to their capacity to reduce systemic oxidative stress. Specific probiotic strains are known to increase the expression of antioxidant enzymes, including those within the thioredoxin pathway. These imply mitigating TXNIP overexpression induced by a high-fat diet, thereby preserving β-cell function and improving hepatic insulin signaling [57,62].

Furthermore, it appears that microbial secondary metabolites such as actinomycin D and rapamycin analogues may be able to alter metabolic signaling [63]. Preliminary study suggests that rapamycin analogues can affect the TXNIP–thioredoxin axis through mTOR signaling, a crucial mechanism that controls cellular stress and survival, even if their involvement in GLP-1 regulation needs more clarification [64]. This points to a possible dual mechanism affecting redox homeostasis and nutrient-sensing pathways.

Still, the microbiota–host interaction within the gut–liver axis has opened new avenues for exploiting microbial biosynthetic pathways in emerging interventions for diseases related to MetS. The current report suggests that the concurrent targeting of GLP-1 secretion and the TXNIP–thioredoxin system by microbial metabolites presents a powerful, holistic strategy to address both the hormonal and oxidative components of MetS.

### 3.4. Defining Criteria for Dual-Target Modulation

Before proceeding to highlight a wide array of natural products with potential effects on GLP-1 signaling or the TXNIP–thioredoxin system, a critical criterion needs to be established for clarity. That is, mechanistic criteria for classifying a natural compound as a genuine dual-target modulator, as opposed to an agent with isolated or coincidental activities on each pathway. Therefore, this subsection aims to delineate these criteria to enhance the rigor and reproducibility of future research.

#### 3.4.1. Inclusion Criteria for Dual-Target Activity

To be considered as a potential dual modulator within the context of this review, a natural product must meet the following evidence-based benchmark. There must be published, peer-reviewed evidence demonstrating a measurable impact on both the GLP-1 pathway and the TXNIP–Thioredoxin system. The former includes, but is not limited to, the stimulation of GLP-1 secretion from enteroendocrine L-cells, inhibition of DPP-4 activity, or direct agonism of the GLP-1R. The latter implies the downregulation of TXNIP expression, enhancement of Trx activity, reduction in associated oxidative stress markers (e.g., ROS), or inhibition of downstream TXNIP-mediated processes, such as NLRP3 inflammasome activation. The supporting evidence may come from a single study that concurrently investigates both mechanisms or from multiple studies which, when considered together, present a compelling case for dual activity. Notedly, the primary goal of these criteria is to shift the narrative from correlative observation to causal, multi-mechanistic investigation.

#### 3.4.2. Differentiation of Evidence Quality: Direct vs. Inferential

It is crucial to distinguish between the quality of data supporting dual modulation and the present state of the research. We therefore propose that researchers should consider a classification system to distinguish compounds mentioned between levels of mechanistic support as:(a)*Direct Evidence*: Compounds with strong, direct evidence from mechanistic investigations confirming simultaneous activity on both targets are assigned to this grade. An excellent illustration is berberine, which has been shown in certain studies to act as a DPP-4 inhibitor while also demonstrating, frequently in the same experimental setup, AMPK-mediated downregulation of TXNIP expression and the resulting reduction in oxidative stress [37,38,65]. This can propose a solid causal connection between the compound’s dual pharmacological properties and its chemistry.(b)*Inferential Evidence*: This group includes substances whose dual action is a tenable, but unproven, deduction derived from a confluence of distinct, well-established activities. Quercetin, for example, is a well-established antioxidant and Nrf2 activator that can alter redox-sensitive pathways such as the TXNIP–thioredoxin system [34]. It has also been experimentally demonstrated to stimulate GLP-1 secretion in L-cells [18,33]. The combined evidence strongly suggests dual regulation; however, to elevate it to the “Direct Evidence” category, definitive research is required that explicitly links its GLP-1 secretagogue action to TXNIP reduction in a single animal.

#### 3.4.3. Importance of Critical Analysis

Implementing these criteria adds a necessary layer of critical analysis to the field. For instance, it can allow researchers to:(a)*Prioritize lead compounds*: That is, to focus resources on compounds with the strongest evidence for genuine polypharmacology, such as berberine or fucoxanthin, for further development.(b)*Identify research gaps*: That is, to highlight promising compounds like quercetin or genistein that require targeted experimental validation to confirm dual-target mechanisms within a single biological system.(c)*Enhance reproducibility*: That is, to encourage standardized experimental designs that assess both incretin and redox endpoints simultaneously. This will lead to more reliable and translatable findings.

In this report, our understanding of how natural products can simultaneously modulate the GLP-1 and TXNIP–thioredoxin axes to achieve a synergistic therapeutic effect in MetS is therefore being advanced from a phenomenological listing of bioactivities to a more accurate and mechanistic understanding.

## 4. Natural Product Mechanisms of GLP-1 Modulation with Potential TXNIP–Thioredoxin System Interconnection

The suggested route, as seen in Figure 2, exemplifies an intricate, multi-layered therapy approach in which natural products simultaneously alter redox homeostasis and hormone signaling to address the intricate pathophysiology of MetS. In order to produce a synergistic therapeutic effect, this method embraces the polypharmacological character of natural substances, going beyond single-target pharmacology.

### 4.1. The Classical Metabolic Pathway of a Natural Product with Dual Action

#### 4.1.1. Initiation: Dual-Pronged Stimulation of Endogenous GLP-1 Secretion

Natural compounds that modulate metabolic pathways originate primarily from two sources: dietary fiber processed by the gut microbiota and directly ingested bioactive phytochemicals. Insoluble dietary fibers resist human digestive enzymes and reach the colon intact, where commensal bacteria, including *Bifidobacterium* and *Lactobacillus*, ferment them into short-chain fatty acids (SCFAs) such as butyrate, propionate, and acetate. Simultaneously, plant-derived compounds like ferulic acid, among phenolic acids, and flavonoids, including quercetin and genistein, are ingested and subsequently absorbed or metabolized within the gastrointestinal tract.

These agents converge on enteroendocrine L-cells in the intestinal epithelium, where they promote GLP-1 release through complementary mechanisms. On the surface of L cells, SCFAs attach to certain G-protein coupled receptors, GPR41 (FFAR3) and GPR43 (FFAR2). Key kinases are activated, and intracellular calcium (Ca^2+^) rises as a result of receptor activation that sets off intracellular signaling cascades involving Gβγ subunits. Concurrently, quercetin and other phytochemicals may enter the L-cell and directly affect these similar pathways, frequently via modifying the activity of kinases like Mitogen-Activated Protein Kinase (MAPK) and Protein Kinase A (PKA).

**Figure 2 antioxidants-14-01364-f002:**
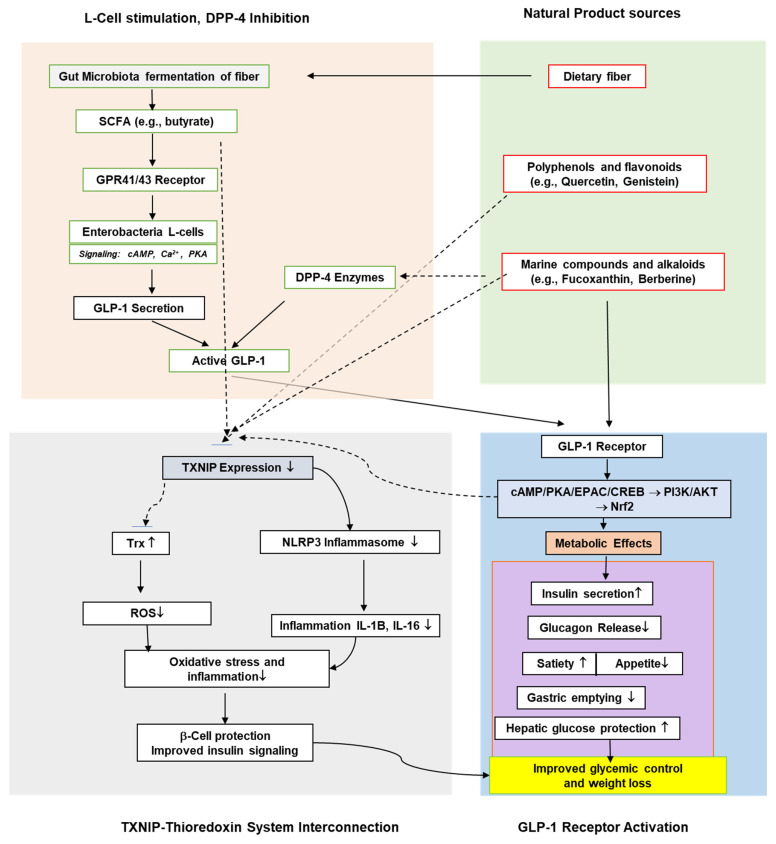
Classical pathway showing the molecular mechanism of natural product targeting GLP-1 modulation with potential TXNIP–Thioredoxin Antioxidant system interconnection. Metabolic effects of GLP-1 receptor activation include PKA/PI3K/AKT Signaling, increased insulin secretion, and satiety with lowered gastric emptying, appetite, glucagon release, and hepatic glucose protection.

The integrated effect of these signaling events triggers the exocytosis of secretory vesicles containing preformed GLP-1, resulting in a physiological elevation of endogenous incretin levels. This mode of enhancement is considered particularly advantageous as it preserves the natural, glucose-dependent pulsatile secretion pattern of GLP-1.

#### 4.1.2. Amplification: Protection of Secreted GLP-1 via DPP-4 Inhibition

A significant drawback of natural GLP-1 is its incredibly brief plasma half-life (1–2 min), which is caused by Dipeptidyl Peptidase-4’s (DPP-4) quick enzymatic destruction. Here, the route uses natural DPP-4 inhibitors, including the alkaloid berberine and sulfated polysaccharides obtained from marine sources, to address this. These compounds are believed to work as competitive or allosteric inhibitors, attaching to the active site of the DPP-4 enzyme and preventing it from cleaving and deactivating GLP-1 [35]. It is noteworthy that this method can prolong the half-life and duration of activity of GLP-1 in the circulation by intensifying the action of what is already released rather than stimulating the synthesis of additional GLP-1 [37]. As a result, it produces a more consistent and long-lasting incretin action than the brief peak caused by secretion alone.

#### 4.1.3. Execution: GLP-1 Receptor Activation and Metabolic Consequences

Through the circulation, the stabilized, active GLP-1 and direct natural GLP-1 receptor agonists (such as fucoxanthin) reach their target organs. A strong intracellular signaling cascade is started when these ligands attach to the GLP-1 Receptor (GLP-1R), a G_S_-protein-coupled receptor. This binding initiates a robust intracellular signaling cascade: adenylate cyclase activation elevates cyclic AMP (cAMP), which in turn activates protein kinase A (PKA) and the exchange protein activated by cAMP (EPAC), ultimately engaging downstream pathways such as PI3K/Akt and MAPK. These signaling events coordinate a range of tissue-specific metabolic responses. In pancreatic β-cells, cAMP/PKA signaling enhances glucose-stimulated insulin secretion while suppressing glucagon release in α-cells; in the brain, GLP-1R activation in the hypothalamus and brainstem promotes satiety; in the stomach, it delays gastric emptying to moderate postprandial glucose levels; and in peripheral tissues including the liver, muscle, and adipose tissue, it improves insulin sensitivity and reduces hepatic glucose production.

The collective outcome becomes a significantly improved glycemic control, promotion of weight loss, and documented cardioprotective benefits.

#### 4.1.4. The Core Synergy: Interconnection with the TXNIP–Thioredoxin System

The direct mechanistic connection between GLP-1 signaling and the control of oxidative stress is the most important and novel feature of this suggested route. Upon GLP-1 receptor activation, the triggered PKA/PI3K/Akt signaling cascades potently suppress TXNIP expression, establishing a core synergistic mechanism as follows. TXNIP serves as a critical redox and metabolic regulator whose expression rises under nutrient excess (e.g., high glucose or free fatty acids), thereby placing a “brake” on the antioxidant response by binding to and inhibiting Trx. GLP-1 signaling effectively “lifts this brake” by downregulating TXNIP, which not only facilitates ROS scavenging and mitigates oxidative damage but also directly restrains NLRP3 inflammasome activation. This may enable the efficient neutralization of ROS and the reduction in oxidative damage. Furthermore, lower TXNIP levels directly inhibit the sparking of the NLRP3 inflammasome, an essential sensor that stimulates the production of pro-inflammatory cytokines like IL-1β that are connected to insulin resistance and β-cell mortality. Together, these effects form a reinforcing feedback loop. The body’s capacity to maintain glycemic control is further supported by increased insulin signaling and β-cell survival and function, which are both influenced by decreased oxidative stress and inflammation.

#### 4.1.5. Amplifying the Synergy: Dual-Targeting by Natural Products and the Concept of Advantageous Polypharmacology

The inherent polypharmacology of natural products, which allows them to operate on several nodes within this network concurrently, greatly increases the therapeutic potential of the integrated GLP-1/TXNIP–thioredoxin pathway. However, it’s important to distinguish between non-specific interactions and advantageous multi-target effects. In this review, “advantageous polypharmacology” is defined as the deliberate and synergistic modulation of several interrelated targets within a disease-related network, such as the simultaneous suppression of the TXNIP–thioredoxin axis and enhancement of GLP-1 signaling, which can result in a consistent and enhanced therapeutic outcome. This contrasts with “non-specific binding” or “off-target toxicity,” which describes inadvertent interactions with physiologically unrelated targets that may result in negative consequences and reduce the effectiveness of treatment. To create natural product-based therapeutics, the ultimate objective is to achieve “selective polypharmacology,” in which the multi-target activities are exactly linked with the pathophysiology of the illness.

Accordingly, the natural product safety profiles, which are frequently influenced by their traditional medical history, might offer early indicators of a good risk-benefit ratio. This historical precedent alone, however, is insufficient; thorough and targeted toxicological screening is necessary, especially when TXNIP suppression is used to modify basic processes like redox homeostasis.

Concrete examples from this pathway that illustrate this principle of advantageous polypharmacology are:(a)Short-chain fatty acids (SCFAs), such as butyrate, operate as histone deacetylase inhibitors (HDACi), which epigenetically suppress the expression of the TXNIP gene, in addition to stimulating GLP-1 secretion by activating G-protein coupled receptors (GPCRs) on L-cells. A synergistic effect on the metabolic network is possible by this twofold, mechanistically separate activity.(b)Flavonoids such as quercetin and genistein promote GLP-1 release while con-currently antagonizing TXNIP expression, often through activation of antioxidant response elements (e.g., via Nrf2 signaling). This direct action on the redox system complements the incretin system, reinforcing the cycle of benefits.(c)Other substances, such as berberine (a DPP-4 inhibitor) and fucoxanthin (a GLP-1 receptor agonist), have been shown to increase total antioxidant capacity, which can downregulate TXNIP via pathways including Nrf2 activation and AMPK, respectively.

Thus, this multi-target action means the beneficial effects on the TXNIP–thioredoxin system are not solely dependent on the cascade initiated by GLP-1 secretion; they could be reinforced directly by the compounds themselves. An excellent illustration of the therapeutic potential of selective, advantageous polypharmacology can be provided by the strong, synergistic mitigation of oxidative stress, which is essential to the progression of MetS.

### 4.2. Therapeutic Implications of the Classical Metabolic Pathway

This integrated pathway presents a paradigm shift in drug development for metabolic diseases—from discrete molecular targeting toward system pharmacology. Natural compounds, with their innate polypharmacology, are perfectly suitable for this strategy. The hormonal deficits, as well as the underlying oxidative stress and inflammation of MetS, are addressed by the concurrent potentiation of the thioredoxin system and increase in GLP-1 signaling. This dual targeting offers several therapeutic advantages: synergistic effects result in improved weight management and glycemic control; enhanced safety by leveraging physiological pathways and compounds with established tolerability; and potential disease modification via β-cell protection and improved insulin sensitivity at a redox level.

As a conceptual innovation of this analysis, the proposed framework provides a mechanistic basis for understanding how natural products can enable a comprehensive, holistic, and potent strategy for managing and preventing MetS and its related comorbidities.

## 5. Evidence from Preclinical Studies of Natural Products Targeting GLP-1 with Potential TXNIP–Thioredoxin System Interconnection

Figure 3 below showcases the preclinical perspectives on the potential of natural products targeting GLP-1 and the potential interconnection with the TXNIP–Thioredoxin system, which are summarized in Table 2.

### 5.1. In Vitro Assays

In vitro studies provide foundational insights into the mechanisms by which natural products influence GLP-1 dynamics and the TXNIP–thioredoxin system at the cellular level [66]. Although with limitations, most commonly, researchers employ enteroendocrine L-cell lines, such as GLUTag and STC-1 cells, to evaluate the GLP-1 secretagogue potential of bioactive compounds [67], while concurrently assessing their impact on oxidative stress markers in pancreatic β-cells, hepatocytes, and adipocytes. For example, quercetin, genistein, and naringenin have been shown to significantly increase GLP-1 secretion in L-cells through pathways involving cyclic adenosine monophosphate (cAMP), protein kinase A (PKA), and extracellular signal-regulated kinase (ERK) [68], with parallel studies demonstrating these same flavonoids suppress TXNIP expression and enhance thioredoxin activity in hepatic and pancreatic cell lines under high-glucose conditions. These assays also help assess calcium influx and activation of transcription factors linked to GLP-1 biosynthesis [69], while additional mechanistic studies evaluate the downstream effects on NLRP3 inflammasome activation and cellular ROS levels. Another in vitro assay target is DPP-4 inhibition, which is frequently assessed using colorimetric or fluorometric enzyme inhibition assays [70]. Newer studies have revealed that several DPP-4 inhibitors also alter the TXNIP–thioredoxin axis. Natural compounds that have been shown to inhibit DPP-4 through either competitive or non-competitive routes include resveratrol, kaempferol, and berberine, depending on their structural characteristics [68,71]. Berberine is especially noteworthy for its dual action of inhibiting DPP-4 and downregulating TXNIP via AMPK activation [71]. These cellular tests are typically used as first dual bioactivity screens, directing further in vivo studies that focus on both the incretin and antioxidant pathways.

### 5.2. In Vivo Animal Models

The physiological effects of natural compounds that dual-modify GLP-1 signaling and the TXNIP–thioredoxin pathway have been confirmed by animal research, especially employing rat models of MetS, type 2 diabetes, or diet-induced obesity [72]. Fucoxanthin, a carotenoid from brown algae, dramatically raised plasma GLP-1 levels, improved glucose tolerance, and decreased body weight in a high-fat diet-induced obese mouse model [72,73]. Subsequent research showed that these benefits were accompanied by noted suppression of hepatic TXNIP expression, increased thioredoxin activity, and decreased systemic oxidative stress markers [16]. AMPK-mediated downregulation of TXNIP and subsequent inhibition of the NLRP3 inflammasome in pancreatic islets were the mechanisms by which berberine, when given orally to diabetic rats, increased endogenous GLP-1 concentrations, improved insulin sensitivity, and suppressed hepatic gluconeogenesis [65,66]. Similarly, in mice with diabetes caused by streptozotocin, ginsenosides from Panax ginseng decreased fasting blood glucose levels and enhanced GLP-1 secretion [74,75]. Through thioredoxin system regulation and a decrease in TXNIP overexpression in kidney and liver tissues, they also showed notable antioxidant benefits. The pharmacokinetic profiles, toxicity data, and dual mechanisms of action that are crucial for translational research are thus provided by these animal studies, which also validate the efficacy seen in vitro. In particular, they show that coordinating the modulation of both the GLP-1 and TXNIP–thioredoxin pathways results in superior metabolic outcomes when compared to targeting either system alone.

**Figure 3 antioxidants-14-01364-f003:**
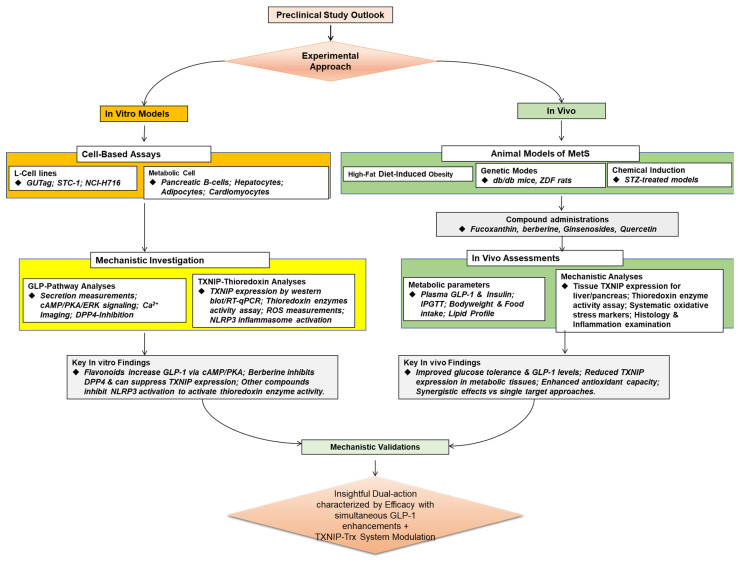
A schematic outlook of the preclinical evidence on the natural products targeting GLP-1 with potential TXNIP–Thioredoxin system interconnection.

**Table 2 antioxidants-14-01364-t002:** Comparative Overview of Effective Doses/Concentrations and Evidence Quality for Natural Products with Dual GLP-1 and TXNIP–Thioredoxin Modulatory Activity.

Natural Product	Source	Primary Mechanism(s) of Action	Effective In Vitro Concentration	Effective In Vivo Dose (Animal Model)	Key Experimental Models	Evidence Type	References
Berberine	*Berberis* species (Alkaloid)	DPP-4 inhibition AMPK-mediated TXNIP suppression via ROS Enhances GLP-1 secretion	10–50 µM (for DPP-4 inhibition & TXNIP downregulation)	50–200 mg/kg/day (oral; HFD/STZ-induced diabetic rats/mice)	STC-1/GLUTag cells Pancreatic β-cell lines HFD/STZ rodent models	Direct: Dual action confirmed in single, integrated studies	[37,38,65,71]
b. Fucoxanthin	Brown Seaweed (Carotenoid)	GLP-1 secretagogue Suppresses hepatic TXNIP expression via ROS Enhances insulin sensitivity	1–10 µM (for GLP-1 secretion & TXNIP suppression in cell lines)	0.1–0.2% dietary supplementation (HFD-induced obese mice)	STC-1 cells Hepatocyte models HFD rodent models	Direct: Dual action confirmed in single, integrated studies	[17,43,44,72,73]
c. Short-Chain Fatty Acids (Butyrate)	Gut Microbiota Fermentation	Activates GPR41/43 on L-cells GLP-1 secretion HDAC inhibition → Epigenetic TXNIP suppression via ROS	0.5–5 mM (for GLP-1 secretion & TXNIP downregulation in colonic cells)	5% (*w*/*w*) in diet or 100–200 mg/kg/day (oral; HFD rodent models)	Colonic epithelial cell line Primary colonic cultures HFD/obesity rodent models	Direct: Dual action confirmed in single, integrated studies	[55,56,59,60]
d. Quercetin	Onions, Apples (Flavonoid)	Stimulates GLP-1 secretion (Ca^2+^/cAMP signaling) Antioxidant; potential TXNIP downregulation via Nrf2	5–20 µM (for GLP-1 secretion in L-cells) 10–50 µM (for antioxidant/TXNIP effects)	25–100 mg/kg/day (oral; HFD-induced obese mice)	STC-1/GLUTag cells HepG2 cells (high glucose) HFD rodent models	Inferential: Separate evidence for each target; dual action in a single system is a reasonable inference.	[18,33,34]
e. Genistein	Soybeans (Isoflavone)	Enhances GLP-1 release Antioxidant; modulates oxidative stress pathways	10–50 µM (for GLP-1 secretion & antioxidant effects)	10–50 mg/kg/day (oral; diabetic rodent models)	Enteroendocrine L-cell lines Pancreatic β-cell lines STZ-induced diabetic models	Inferential: Separate evidence for each target; dual action in a single system is a plausible inference.	[35,36]
f. Epigallocatechin Gallate (EGCG)	Green Tea (Polyphenol)	DPP-4 inhibition Antioxidant; suppresses TXNIP/NLRP3 inflammasome	10–100 µM (for DPP-4 inhibition & anti-inflammatory effects)	50–100 mg/kg/day (oral; db/db mice, HFD models)	Enzyme inhibition assays Macrophage/β-cell inflammation models Genetic & diet-induced diabetic models	Inferential: Separate evidence for each target; dual action in a single system is a plausible inference.	[39,41]
g. Ginsenosides	*Panax ginseng* (Saponins)	Stimulates GLP-1 secretion Antioxidant; reduces TXNIP overexpression in tissues.	10–50 µg/mL (for GLP-1 secretion & cytoprotection)	100–200 mg/kg/day (oral; STZ-induced diabetic mice)	STC-1 cells Pancreatic islet studies STZ-induced diabetic models	Inferential: Separate evidence for each target; dual action in a single system is a plausible inference.	[69,75]

## 6. Pharmacological Challenges and Opportunities for Natural Products Targeting GLP-1 with TXNIP–Thioredoxin System Interconnection

Natural GLP-1-modulating molecules face several pharmacological and developmental challenges in the development of effective therapies, despite promising preclinical findings. These obstacles become more complex when considering that these compounds target both the GLP-1 and TXNIP–thioredoxin systems. To advance natural products as efficient, multi-mechanistic therapies for disorders associated with MetS, these obstacles must be addressed (see Figure 4).

### 6.1. Bioavailability, Metabolic Stability, and Dose–Response Considerations

The low metabolic stability and poor oral bioavailability of many natural substances are among the main obstacles to natural products. Because of their low solubility, acidic instability, and fast hepatic metabolism, phytochemicals such as quercetin and curcumin have little systemic availability in vivo while having strong bioactivity in vitro [76,77,78]. For dual-target therapies, as seen in Table 2, this is especially important since they need enough chemical levels to reach intestinal L-cells (to promote GLP-1 secretion) as well as systemic metabolic regions such as the liver and pancreas (to exhibit antioxidant effects via TXNIP suppression). The quantities needed for TXNIP suppression in vitro (10–50 μM for berberine, for example) are sometimes hard to get systemically with oral dosing, indicating a major pharmacokinetic barrier. To overcome this and close the gap between in vitro efficacy and in vivo applicability, formulation innovations must be strategic and include the use of bioenhancers, liposomes, micelles, and polymeric nanoparticles to boost absorption and target tissue delivery [79,80].

### 6.2. Standardization, Reproducibility, and Toxicity Profiling

Reproducibility and standardization are significantly hampered by the intrinsic heterogeneity in the chemical composition of natural products, which depends on the source species, extraction technique, and growth circumstances. This diversity represents a substantial translational gap and makes efficacy comparisons more difficult. Future research must use chemically characterized extracts with measured active ingredients while using Good Agricultural and Collection Practices (GACP) to guarantee consistency [14]. Furthermore, because changing fundamental systems like the TXNIP–thioredoxin axis has a profound pharmacological impact, extensive, long-term toxicological research is necessary even if natural chemicals are believed to be harmless. To create a favourable risk-benefit ratio for these multi-target drugs, meticulous dose–response studies and specialized safety profiles are necessary due to the possibility of unexpected effects from persistent TXNIP suppression [81].

### 6.3. Delivery Challenges: Formulation Strategies for Oral vs. Injectable Routes

Many bioactive natural chemicals are peptide-like or labile, which makes distribution extremely difficult. Alternative approaches are frequently required due to the GI tract’s enzymatic breakdown, even though oral delivery is recommended for patient compliance. Research must concentrate on creating prodrug strategies, oral peptide technologies that use enzyme inhibitors or permeation enhancers, and gastro-resistant formulations in order to get around this [82,83]. The creation of delivery methods that target the colon may be especially beneficial since it would enable localized effect on the gut microbiota for the synthesis of SCFAs like butyrate, which are strong TXNIP inhibitors, as well as L-cells for GLP-1 secretion [84]. A significant breakthrough in the development of non-injectable, patient-friendly dual-target treatments would result from success in this field.

### 6.4. Regulatory and Development Pathways for Multi-Target Agents

Compared to single-target synthetic medications, the regulatory environment for multi-target natural products is still complicated and unstructured. Interestingly, many substances are excluded from the stringent clinical trial procedures needed for pharmaceutical clearance since they are classified as herbal supplements or nutraceuticals. Clearer regulatory frameworks that recognize the polypharmacological character of these medicines are necessary to close this translational gap. To provide a feasible path for clinical translation and integration into standard care, this involves developing pathways for combination therapies where natural products function as adjuvants to currently available pharmaceuticals and standardizing testing protocols for dual endpoints (metabolic and oxidative stress markers) [85,86].

Therefore, despite the significant pharmacological potential of natural GLP-1 modulators that also target the TXNIP–thioredoxin pathway, their effective development necessitates overcoming significant regulatory and scientific obstacles. Together with specific regulatory pathways, strategic advancements in formulation, standardization, toxicity screening, and administration methods can aid in the conversion of these promising, multi-target natural chemicals into clinically applicable treatments that target the oxidative and hormonal cores of MetS.

## 7. Integrative and Computational Approaches in Natural Products Targeting GLP-1 with TXNIP–Thioredoxin System Interconnection

Recent advances in computational biology and systems pharmacology provide powerful, albeit evolving, tools for discovering and characterizing natural products that may dually target the GLP-1 pathway and the TXNIP–thioredoxin system. These integrative approaches are useful for streamlining early-phase drug discovery, predicting multi-pharmacological targets, and generating hypotheses about complex, synergistic mechanisms of action [18,87]. However, it is critical to assess these methods with a clear understanding of their current predictive power and validation status. The workflow for this integrated computational approach is illustrated in Figure 5.

### 7.1. Role of Molecular Docking in Multi-Target Prediction: From Structure to Hypothesis

Molecular docking has become a standard tool for investigating interactions between natural compounds and key protein targets. As reviewed by Agu et al. [88], simulating the binding of ligands to receptors like GLP-1R, DPP-4, and TXNIP provides initial insights into binding affinity and orientation. Docking studies can be utilized in dual-target discovery to anticipate how a chemical may, for example, bind to the thioredoxin-interaction site of TXNIP and inhibit DPP-4 [60,89]. These forecasts are useful for formulating hypotheses and ranking lead compounds for validation through experimentation. It is crucial to remember that, whereas docking may reliably forecast binding modes for single targets—retrospective studies have shown that effective validation rates for targets like DPP-4 are very high—its predictive ability for genuinely unique, dual-target mechanisms is less well-established [90,91]. Therefore, positive docking results must be regarded as preliminary data that require thorough biochemical and cellular validation, and the experimental success rate for completely novel dual-target predictions is still a topic of active research [92].

**Figure 5 antioxidants-14-01364-f005:**
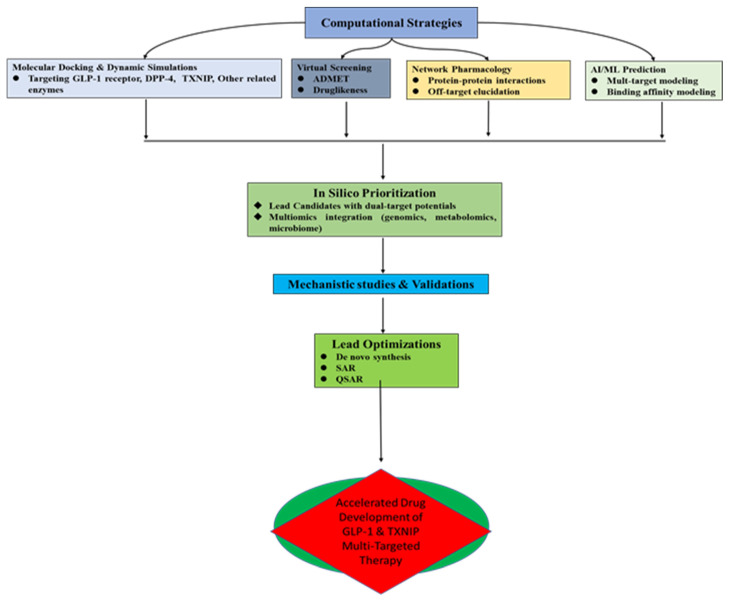
Flow diagram of integrated computational approach towards accelerated GLP-1 and TXNIP dual target therapeutic discovery.

### 7.2. Use of Network Pharmacology to Identify Interconnected Pathway Links

Network pharmacology connects bioactive substances to many gene and protein targets within disease pathways, allowing for a comprehensive investigation of these chemicals. This method is ideal for developing theories at the systems level about the co-regulation of the GLP-1 and TXNIP–thioredoxin axis by natural products [87]. According to network analysis, for instance, berberine or quercetin may influence Nrf2-mediated antioxidant responses and operate on larger networks that link GLP-1 secretion to downstream TXNIP suppression through PKA/Akt signaling [84,93]. Researchers can find key “hub” targets and possible synergistic processes by combining data on protein–protein interactions with compound–target networks. This method’s strength is its capacity to produce thorough mechanistic hypotheses from available data. However, the predictions made by network pharmacology are necessarily correlative and should not be considered definitive proof of efficacy, but rather as a basis for designing experiments. The capacity of these models to provide testable hypotheses that are subsequently confirmed—a process that necessitates focused in vitro and in vivo validation—is used to gauge their success rate [84,94].

### 7.3. In Silico Screening of Natural Product Libraries: Balancing Throughput with Validation

Finding possible dual modulators can be sped up by virtual screening of sizable natural product libraries. Researchers can test hundreds of chemicals against various targets using databases such as ZINC and NPASS [95]. Cost-effective filtering based on anticipated bioactivity and drug-likeness is made possible by these high-throughput techniques. Gomaa et al. [89], for example, showed how to employ these techniques to find non-peptidic GLP-1R modulators. Virtual screening’s main benefit is that it may quickly reduce the number of candidates from thousands to a manageable number of “hits” for experimental testing, which drastically lowers resource costs. However, a significant limitation is the rate of false positives; not all computationally predicted hits will show activity in biological assays. For example, the experimental success rate can vary widely depending on the quality of the library, the accuracy of the target structures, and the stringency of the filtering criteria [95]. Therefore, hits from in silico screening must be considered as promising leads for further investigation, not as confirmed active compounds.

### 7.4. Application of AI in Multi-Target Natural Drug Discovery Pipelines: A Promising Frontier

Artificial intelligence (AI) and machine learning are transforming drug discovery by analyzing complex, multi-dimensional datasets. AI can predict novel compound–target interactions by learning from cheminformatic, genomic, and metabolomic data [96]. For dual targeting, AI models can be trained to identify structural features associated with, for example, both DPP-4 inhibition and thioredoxin system enhancement [97]. This approach holds great promise for de novo drug design and multi-omics integration. However, it is crucial to frame this as a promising but still-emerging approach. The application of AI specifically for predicting dual GLP-1/TXNIP modulation is in its infancy. The robustness of such models is highly dependent on the quality and quantity of available training data, which, for this specific dual mechanism, is currently limited. While AI excels at pattern recognition, its predictions for complex polypharmacology must be subjected to the same rigorous experimental validation as other computational methods. Its current greatest utility is in augmenting and accelerating the hypothesis-generation phase of the drug discovery pipeline [98].

Thus, integrative and computational tools can provide a powerful platform for advancing natural therapeutics that co-target the GLP-1 and TXNIP–thioredoxin systems. They are best viewed as sophisticated methods for generating testable hypotheses and prioritizing experiments. These computational techniques can successfully supplement conventional pharmacological methods by recognizing their limits and highlighting the critical requirement for experimental validation. This will offer up new possibilities for MetS precision drug discovery.

## 8. Implications of Targeting GLP-1 with TXNIP-Thioredoxin System for Drug Discovery and Clinical Translation

In the current research mainstream, the considerable therapeutic potential of natural products is phenomenal in the treatment of MetS and its related complications, as highlighted by the rising interest in them as modulators of GLP-1 signaling and the TXNIP–thioredoxin system. Notably, natural substances uphold unique benefits that complement current trends in metabolic health interventions as the pharmaceutical industry shifts towards safer, more affordable, and multi-targeted treatments that address both hormonal imbalance and underlying oxidative stress. Accordingly, Figure 6 elucidates the route towards implications for clinical translation.

### 8.1. Addressing the Translational Gap: From Animal Models to Human Trials

There is a substantial translational gap between human clinical validation and encouraging animal research. Although rodent models of MetS have been useful, there hasn’t been much progress towards human trials, especially for research looking at both incretin and antioxidant outcomes at the same time. Future clinical trials must be planned with well-defined dual outcomes to close this gap. In addition to standard glycemic parameters (e.g., HbA1c, fasting glucose, and plasma GLP-1 levels), these should also include direct biomarkers of target engagement for the TXNIP–thioredoxin system, such as thioredoxin activity, plasma TXNIP levels, or indicators of systemic oxidative stress (e.g., isoprostanes) [18,19]. Confirming the dual mechanism of action in humans and proving a direct causal relationship between therapy and effectiveness requires this biomarker-driven approach.

### 8.2. Rationale for Combination Therapies with Existing Agonists

Using natural products as adjuvants to existing GLP-1 receptor agonists is a potential translational technique that solves the shortcomings of current GLP-1 medicines. For instance, a synthetic GLP-1 agonist’s half-life might be extended by co-administering a natural DPP-4 inhibitor [99]. This could minimize adverse effects such as gastrointestinal distress and perhaps allow for a dosage decrease. More significantly, by lowering the underlying oxidative stress and inflammation that lead to GLP-1 resistance and β-cell dysfunction, a natural substance that also inhibits TXNIP may offer further advantages. This synergistic approach may increase the efficacy of traditional therapy while also providing organ-protective advantages not currently availed by monotherapies.

### 8.3. Prioritizing Bioavailability and Formulation in Clinical Design

As stressed, resolving these drugs’ bioavailability issues is crucial to their clinical translation. Therefore, to evaluate the bioavailability of novel formulations (such as nano-encapsulated curcumin or berberine), pharmacokinetic studies should be conducted in conjunction with early-phase human trials. As per Alum et al. [100], a crucial first step in confirming the dual-target approach’s clinical applicability is proving that therapeutic concentrations can be reached in human plasma and tissues. Making formulation science a top priority from the start is essential for clinical success and goes beyond simple optimization.

## 9. Future Research Priorities and Recommendations

A coordinated strategy approach across many crucial fronts is necessary to advance natural product research for multi-targeted drug development in MetS, with an emphasis on the interrelated GLP-1 and TXNIP–thioredoxin systems. In order to identify the precise molecular pathways—such as GLP-1-mediated cAMP/PKA signaling versus GLP-1R-independent pathways like direct Nrf2 activation or epigenetic modulation via HDAC inhibition—through which natural compounds simultaneously modulate incretin signaling and suppress TXNIP to enhance thioredoxin activity, future research must prioritize thorough mechanistic studies that go beyond phenomenological observations. To establish causal relationships, these studies must use cutting-edge tools like CRISPR-Cas9 gene editing in GLP-1R [101] or TXNIP knockout [102] models. Dual bioactivity profiling that includes both redox-specific metrics (TXNIP expression, thioredoxin activity, NLRP3 inflammasome inhibition) and incretin-related endpoints (GLP-1 secretion, DPP-4 inhibition) must be developed in tandem with this to guarantee chemical reproducibility. Furthermore, breakthroughs in formulation science, such as prodrug technologies and nanoencapsulation, are necessary to overcome bioavailability limitations. These developments are intended to guarantee targeted distribution to intestinal L-cells as well as systemic metabolic regions like the liver and pancreas. While the clinical translation pipeline needs to be strengthened through biomarker-driven human trials that incorporate direct measures of oxidative stress and inflammation, sustained by long-term pharmacovigilance and regulatory harmonization to validate the real-world efficacy and safety of these multi-target therapeutic strategies, the integration of advanced computational tools and artificial intelligence is essential for speeding up the virtual screening of compound libraries for dual affinity and creating predictive models of polypharmacology [100]. Admittedly, achieving this research goal is an ambitious agenda that needs funding and an ethical framework. Considering the prevailing lack of economic interest and intellectual property protection, researchers need to build linkages through Public and Philanthropic Grants, Public–Private Partnerships, and Regulatory Incentives.

## 10. Conclusions

This comprehensive review has piqued the growing interest in pharmaceutical research towards leveraging natural products as multi-target modulators of both GLP-1 signaling and the TXNIP–thioredoxin system for the comprehensive treatment of MetS and its associated conditions. As observed from existing reports, preclinical evidence demonstrates that various plant-derived compounds, marine bioactives, and microbial metabolites can not only stimulate GLP-1 secretion, inhibit DPP-4 activity, or mimic GLP-1 receptor agonism but also concurrently suppress TXNIP expression and enhance thioredoxin-mediated antioxidant defense, offering a powerful dual-mechanistic approach for metabolic and redox regulation. Agreeably, these reports reinforce the considerable therapeutic potential of natural products as viable alternatives or adjuncts to current pharmacologic treatments by simultaneously addressing the incretin deficiency and chronic oxidative stress that underpin MetS, thereby potentially enhancing efficacy while improving patient compliance and reducing adverse effects. However, despite substantial in vitro and in vivo data supporting this dual-axis modulation, a major translational gap remains in advancing these discoveries into clinical applications, hindered by challenges in bioavailability, standardization, and mechanistic clarity for polypharmacological agents. Therefore, future research must focus on rigorous clinical validation incorporating biomarkers of both metabolic and oxidative stress responses, standardized formulation development using advanced delivery systems to ensure target tissue engagement, and integrative approaches using advanced computational tools and AI to optimize discovery, delivery, and regulatory readiness for these multi-target therapeutics. To fully use the synergistic benefits of natural products in the upcoming generation of treatments that simultaneously target GLP-1 pathways and the TXNIP–thioredoxin system for holistic metabolic health management, such a focused research emphasis will undoubtedly be necessary.

## Figures and Tables

**Figure 1 antioxidants-14-01364-f001:**
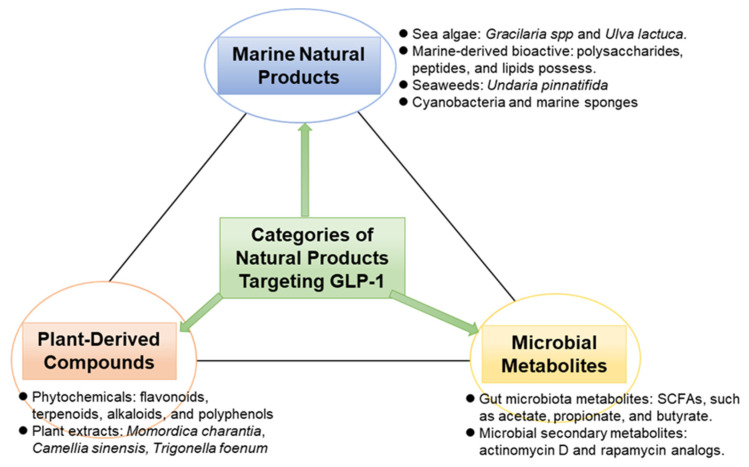
Classifications of natural products targeting GLP-1.

**Figure 4 antioxidants-14-01364-f004:**
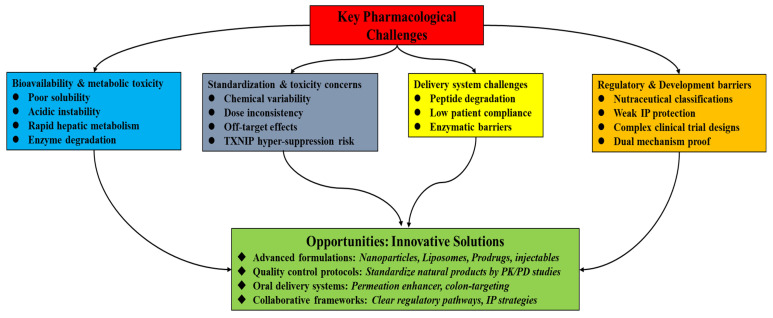
A layout of several pharmacological and developmental challenges and opportunities in the development of effective natural product-based therapies with dual-target modulatory actions on GLP-1 and TXNIP.

**Figure 6 antioxidants-14-01364-f006:**
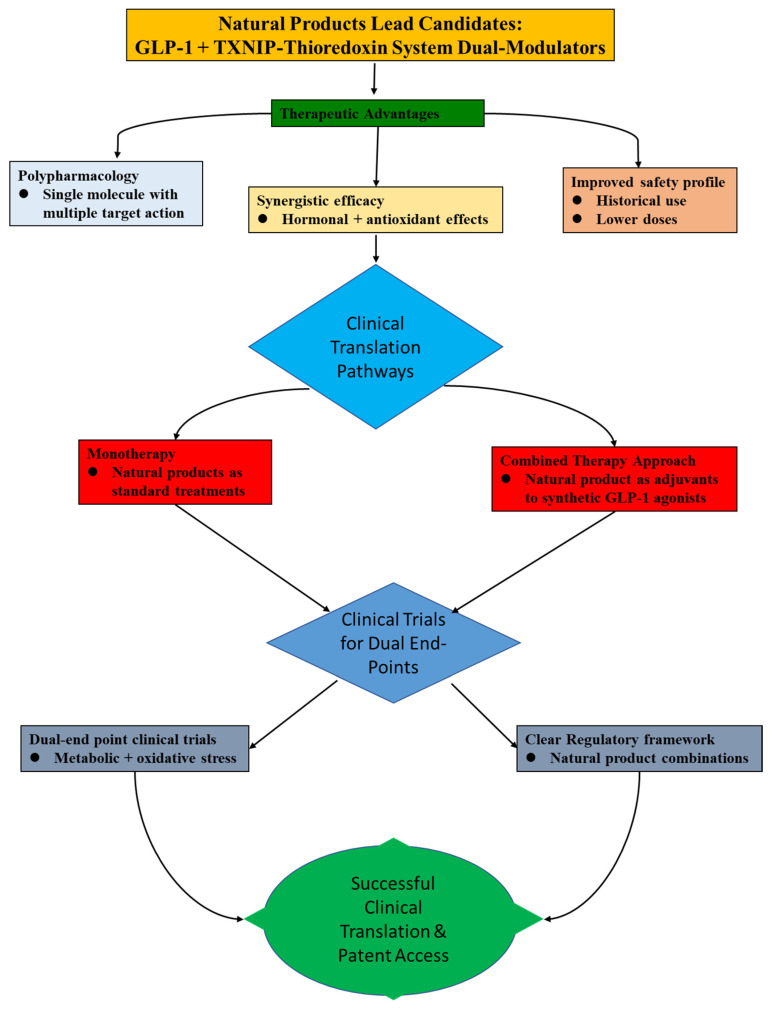
Showing the clinical translation pathways for natural product lead candidates that act as dual modulators of the GLP-1 and TXNIP–Thioredoxin systems.

**Table 1 antioxidants-14-01364-t001:** Summary on GLP-1 with TXNIP–Thioredoxin System Interconnection in MetS.

Aspect	GLP-1 System	Thioredoxin Antioxidant System
Primary Role	Hormonal regulator of glucose metabolism, insulin secretion, and satiety.	It is a key cellular defense system that maintains redox homeostasis and reduces oxidative stress.
Key Components	GLP-1 hormone GLP-1 Receptor (GLP-1R) DPP-4 enzyme	Thioredoxin (Trx) Thioredoxin Reductase (TrxR) Thioredoxin-Interacting Protein (TXNIP)
Mechanism of Action	Binds to GLP-1R Stimulates glucose-dependent insulin secretion Suppresses glucagon release Slows gastric emptying Promotes satiety in the brain	Trx reduces oxidized proteins and scavenges ROS. TrxR uses NADPH to recycle Trx to its active state. TXNIP binds to and inhibits Trx, amplifying oxidative stress.
Dysregulation in MetS	*Incretin Deficiency*: Reduced secretion by L-cells. *Short Half-life*: Rapid degradation by DPP-4.	*Oxidative Stress*: Overproduction of ROS. *TXNIP Upregulation:* Increased by high glucose/ROS, leading to Trx inhibition.
Consequences of Dysregulation	Postprandial hyperglycemia Hyperglucagonemia Weight gain	Insulin resistance (via stress kinases) β-cell dysfunction and apoptosis Inflammation (via NLRP3 inflammasome) Vascular dysfunction
Current Therapeutics	GLP-1 Receptor Agonists (e.g., liraglutide, semaglutide) DPP-4 Inhibitors (e.g., sitagliptin)	No direct TXNIP inhibitors or Trx system enhancers are in clinical use; it is an emerging drug target.
Limitations of Synthetics	Injectable administration GI side effects (nausea, vomiting) High cost Long-term adherence challenges	N/A
Therapeutic Goal	Enhance GLP-1 signaling to improve glycemic control, promote weight loss, and protect β-cells.	Inhibit TXNIP and enhance the Trx system to reduce oxidative stress, inflammation, and cellular damage.
Interconnection	GLP-1 signaling enhances the antioxidant function of the Thioredoxin system and provides cytoprotection by downregulating TXNIP expression (via the cAMP/PKA & PI3K/Akt pathways) [15,30,31,32]. The phosphorylation of transcription factors (e.g., CREB) suppresses TXNIP gene transcription, thereby inhibiting TXNIP. With this inhibition of TXNIP, GLP-1 signaling may increase antioxidant defence and reduce inflammation. This, in turn, improves β-cell function and insulin sensitivity, leading to improved glycemic control and reduced metabolic stress [31]. The multifaceted effects of natural products, which operate on both axes, show that this produces a stronger and more persistent therapeutic impact than monotherapy [15].	

## Data Availability

This study does not involve any new data; hence, a data declaration statement is not applicable.

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
