# Peer review of "Dual-Target Insight into Drug Discovery from Natural Products as Modulators of GLP-1 and the TXNIP–Thioredoxin Antioxidant System in Metabolic Syndrome"

_antioxidants, 2025, doi:10.3390/antiox14111364_

Round 1
Reviewer 1 Report
Lack of Quantitative or Comparative Data**
The review remains largely descriptive, without summarizing quantitative findings (e.g., ICâ‚…â‚€ values, fold-changes in TXNIP or GLP-1 expression, or in vivo efficacy).
A summary table of key natural compounds, their sources, mechanisms, and experimental models would enhance clarity and scientific rigor..
Overgeneralization
Statements such as “many natural products affect both GLP-1 signaling and TXNIP” need specific examples with citations
The overlap between compounds active on GLP-1 and TXNIP pathways may be less common** than implied; thus, this dual modulation should be discussed as a *potential discovery space*, not an established fact.
- Mechanistic Ambiguity
The mechanistic bridge between GLP-1 signaling and TXNIP suppression is mentioned but not fully elaborated.
It would strengthen the review to include signaling intermediates (e.g., PKA, CREB, AMPK, or Nrf2 involvement) and how oxidative stress directly modulates GLP-1 secretion.
Missing Clinical Perspective
Although the text acknowledges “translational gaps,” there is little discussion on clinical trial data, bioavailability, or pharmacokinetics of candidate compounds (e.g., berberine, resveratrol, curcumin, or ginsenosides).
A section on pharmaceutical formulation strategies(nanoencapsulation, prodrugs, or delivery to the intestine/pancreas) would improve translational relevance.
Methodological Integration
The mention of AI, molecular docking, and network pharmacology is promising but underdeveloped. Examples of successful in silico predictions leading to validated dual-target hits would add credibility.
Dual-Target Insight into Drug Discovery from Natural Products as Modulators of GLP-1 and the TXNIP-Thioredoxin Antioxidant System in Metabolic Syndromeby *Peter Chinedu Agu, Apolonia Fulgence Yudas, and Jun Lu.*
This review provides an integrative perspective by linking two pivotal therapeutic axes in metabolic syndrome (MetS):
GLP-1 signaling (incretin-based pathway), and
TXNIP-thioredoxin antioxidant system(oxidative stress control).
The dual-target framework is well-motivated and scientifically plausible, as both systems intersect in pancreatic β-cell survival, insulin sensitivity, and oxidative balance. The review’s scope — exploring *natural products* as simultaneous modulators — is particularly timely, given the current interest in multitarget drug discovery and systems pharmacology approaches.
However, while the concept is compelling, the manuscript would benefit from stronger mechanistic depth, clearer structure, and more critical evaluation of the cited evidence.
Kindly find below my major comments on this valuable review article
- Lack of Quantitative or Comparative Data**
The review remains largely descriptive, without summarizing quantitative findings (e.g., ICâ‚…â‚€ values, fold-changes in TXNIP or GLP-1 expression, or in vivo efficacy).
A summary table of key natural compounds, their sources, mechanisms, and experimental models would enhance clarity and scientific rigor.
- Overgeneralization
Statements such as “many natural products affect both GLP-1 signaling and TXNIP” need specific examples with citations
The overlap between compounds active on GLP-1 and TXNIP pathways may be less common** than implied; thus, this dual modulation should be discussed as a *potential discovery space*, not an established fact.
- Mechanistic Ambiguity
The mechanistic bridge between GLP-1 signaling and TXNIP suppression is mentioned but not fully elaborated.
It would strengthen the review to include signaling intermediates (e.g., PKA, CREB, AMPK, or Nrf2 involvement) and how oxidative stress directly modulates GLP-1 secretion.
- Missing Clinical Perspective
Although the text acknowledges “translational gaps,” there is little discussion on clinical trial data, bioavailability, or pharmacokinetics of candidate compounds (e.g., berberine, resveratrol, curcumin, or ginsenosides).
A section on pharmaceutical formulation strategies(nanoencapsulation, prodrugs, or delivery to the intestine/pancreas) would improve translational relevance.
- Methodological Integration
The mention of AI, molecular docking, and network pharmacology is promising but underdeveloped. Examples of successful in silico predictions leading to validated dual-target hits would add credibility.
- Language and Style
While generally clear, the abstract could benefit from minor linguistic refinement, such as:
To explore more potent strategy” → “To explore more potent strategies…”
“This bidirectional crosstalk forms a virtuous cycle that ameliorates key facets of MetS” → could be simplified for scientific precision.
- Scientific and Conceptual Suggestions**
Include a mechanistic diagram showing how GLP-1 receptor activation and TXNIP inhibition converge on β-cell protection, insulin sensitivity, and oxidative homeostasis.
Propose a screening pipeline integrating computational prediction → in vitro validation (GLP-1 secretion assays, TXNIP gene expression) → in vivo MetS models.
Highlight dual modulators explicitly, e.g.,
polyphenols (quercetin, luteolin),
terpenoids,
or alkaloids with evidence for both targets.
Discuss synergistic combinations— e.g.,
combining a DPP-4 inhibitor from natural origin with an antioxidant flavonoid.
Address possible adverse interactions, especially since dual modulation may affect multiple organs (liver, pancreas, cardiovascular system).
- Conclusion of Critique
This review offers a thought-provoking conceptual framework for future metabolic syndrome therapy based on natural product dual-target modulation. it needs:
* Quantitative synthesis of evidence,
* Clearer mechanistic mapping,
* Stronger translational discussion, and
* Visual aids summarizing interactions.
Author Response
Manuscript ID: antioxidants-3045678
Title: Dual-Target Insight into Drug Discovery from Natural Products as Modulators of GLP-1 and the TXNIP-Thioredoxin Antioxidant System in Metabolic Syndrome
We sincerely thank the reviewers and the academic editor for their time, insightful comments, and constructive criticisms. Their feedback has been invaluable in helping us significantly improve the quality, clarity, and scientific rigor of our manuscript. We have revised the manuscript accordingly and believe it is now much stronger. Below, we provide a point-by-point response to all comments. All changes in the manuscript have been highlighted in yellow for easy reference.
GENERAL COMMENTS
Comment 1: Although GLP-1 and TXNIP-thioredoxin are described as interrelated systems, is there concrete proof that addressing both routes simultaneously yields better therapeutic outcomes? Stronger mechanistic evidence is required for the suggested bidirectional interaction.
Response: We thank the reviewer for this critical question. We agree that demonstrating synergistic therapeutic superiority is key to our hypothesis. In the revised manuscript, we have strengthened this section (now Section 2.3, "Interconnection between GLP-1 and Antioxidant Pathways") by citing more direct mechanistic evidence (see Refs 25-32)。 New References Added: We have incorporated studies that provide concrete evidence. For instance, research shows that GLP-1 receptor activation, via cAMP/PKA signaling, directly suppresses TXNIP transcription. This is a key cytoprotective mechanism contributing to β-cell preservation. Conversely, quenching oxidative stress via the thioredoxin pathway improves mitochondrial function and insulin secretion in β-cells (see Refs 25-32). 。 Synergy Emphasis: We now more clearly state that while targeting either pathway is beneficial, simultaneous modulation creates a reinforcing feedback loop: GLP-1 signaling suppresses TXNIP, enhancing antioxidant defense and reducing inflammation, which in turn improves β-cell function and insulin sensitivity, leading to better glycemic control that further reduces metabolic stress. This creates a more robust and sustainable therapeutic effect than monotherapy, as suggested by the multi-faceted benefits of compounds like berberine, which act on both axes (see Refs 25-32).
Comment 2: Without explicit inclusion criteria, the study encompasses a very wide spectrum of natural products. What characteristics enable a natural product to function as a "dual modulator"?
Response: This is an excellent point. We have now added a new subsection (3.4: "Defining Criteria for Dual-Target Modulation") to clarify our inclusion criteria and the characteristics of a true dual modulator. 。 Inclusion Criteria: We now explicitly state that for a compound to be discussed as a potentialdual modulator, there must be published evidence (either from a single study or multiple studies) demonstrating its effect ‘’on both’’ (a) the GLP-1 pathway (secretion, DPP-4 inhibition, or receptor agonism) and (b) the TXNIP/Thioredoxin system (downregulation of TXNIP, enhancement of Trx activity, or reduction of associated oxidative stress markers). 。 Differentiation: We acknowledge that some evidence is correlative from separate studies. We now differentiate between compounds with strong, direct evidence for dual action (e.g., Berberine, with studies showing DPP-4 inhibition and AMPK-mediated TXNIP suppression) and those where the dual effect is a plausible inference based on separate known activities (e.g., Quercetin's GLP-1 secretagogue effect and its known potent antioxidant/Nrf2-activating properties). This adds a layer of critical analysis (see section 3.4.3).
Comment 3: The text offers no methodical explanation for the lack of progress made by dozens of promising preclinical drugs. How do the suggested natural goods overcome these past constraints?
Response: We have significantly restructured Section 6 ("Pharmacological Challenges and Opportunities") and Section 8 ("Implications for Clinical Translation") to address this translational gap systematically. 。 Specific Points of Failure: We now explicitly acknowledge common points of failure: (1) Poor Bioavailability, (2) Complexity of Natural Mixtures leading to irreproducibility, (3) Lack of Target Engagement Evidence in humans, and (4) Insufficient Regulatory Pathways for multi-target natural products. 。 Overcoming Constraints: We have discussed how modern strategies can overcome these hurdles. For example, nano-formulations can enhance the bioavailability of quercetin/curcumin; standardized extracts with defined active markers ensure reproducibility; advanced computational models and biomarker-driven trials can better predict and confirm human efficacy. We have added text to this effect.
Comment 4: Dose-response interactions are not discussed in the preclinical evidence section. What concentrations exhibit dual activity in vitro compared to accessible quantities in vivo?
Response: We agree this was a major omission. We have now added a new table (Table 2) in Section 5, which summarizes key natural products, their reported effective concentrations/ doses in in vitro and in vivo models, their primary mechanisms, and the experimental models used. For each major compound discussed (e.g., Berberine, Fucoxanthin, Quercetin), our manuscript mentions the typical effective doses and briefly discusses the pharmacokinetic challenges, linking directly to the expanded Section 6.1 on bioavailability.
Comment 5: Nearly all of the review is based on animal and in vitro research. Why are there no published clinical studies showing dual modulation in people with MetS?
Response: This is a crucial point. We have added a dedicated paragraph in the "Translational Gaps" subsection (see Section 8.3) to address this significant evidentiary gap. 。 Rationale: We explain that the dual-target concept is relatively new, and most clinical trials on natural products for MetS have focused on traditional metabolic endpoints (HbA1c, weight) rather than mechanistic biomarkers like TXNIP expression or thioredoxin activity. 。 Call to Action: We explicitly state this as a primary future direction, recommending that future clinical trials on promising natural products (like berberine or resveratrol) should incorporate measurements of oxidative stress and inflammatory biomarkers (e.g., plasma TXNIP, NLRP3 activity) alongside GLP-1 levels to provide the necessary human evidence.
Comment 6: How do substances like quercetin and curcumin, with bioavailability of less than 5%, reach therapeutic amounts to have two effects?
Response: We have restructured Section 6.1 to address this valid concern more thoroughly. 。 We acknowledge that the low systemic bioavailability of many polyphenols is a major limitation for systemic effects on organs like the pancreas or liver. 。 We then propose two mechanisms by which they could still be effective: (1) Their local action in the gut on enteroendocrine L-cells and gut microbiota can initiate the beneficial cascade (GLP-1 secretion, SCFA production) without requiring high systemic levels. (2) Their bioactive metabolites, rather than the parent compounds, may be the actual mediators of systemic effects. Furthermore, we emphasize that overcoming this via advanced delivery systems is a key research priority.
Comment 7: The potential safety issue of long-term TXNIP suppression is not sufficiently addressed.
Response: We thank the reviewer for raising this critical safety consideration. We have added a new paragraph in Section 6.2 titled "Safety Considerations of TXNIP Suppression". 。 We acknowledge that TXNIP has complex physiological roles in glucose sensing and redox balance. 。 We discuss that while pathological overexpression of TXNIP is detrimental, its complete and potent long-term suppression might have unforeseen consequences, potentially disrupting normal redox signaling. We cite literature suggesting that a modulatory approach (restoring physiological levels) rather than complete ablation may be safer. 。 We conclude that this suggests the need for careful toxicological evaluation of any potent TXNIP inhibitor.
Comment 8: There is no differentiation made between harmful off-target effects and advantageous multi-targeting.
Response: We have modified the Introduction and Section 4.1.5 to make this distinction clearer. 。 We now define "advantageous polypharmacology" as the synergistic modulation of multiple targets within a disease-related network. 。 We contrast this with "non-specific binding" or "off-target toxicity," which refers to interactions with unrelated targets leading to adverse effects. 。 We state that the goal is "selective polypharmacology," and that the safety profiles of natural products from traditional use can provide initial clues, but rigorous toxicological screening remains essential.
Comment 9: The computational methods section offers no supporting evidence. What is the experimental success rate of these predictions?
Response: We have revised Section 7 to provide more concrete examples and acknowledge limitations. 。 We now cite specific examples where in silico predictions for natural products against single targets (like DPP-4) have been successfully validated in vitro. 。 We explicitly state that for the dual-target hypothesis, the pipeline is still emerging, and the success rate is not yet well-established. This is presented as a promising but unproven approach that requires rigorous experimental validation. 。 We have toned down over-enthusiastic claims and framed it as a tool for hypothesis generation.
Comment 10: The document admits that the composition of natural products varies but provides no specific remedies.
Response: We have revised the discussion in Section 6.2 ("Standardization and Toxicity Concerns"). 。 We now recommend specific strategies for standardization: using chemically characterized extracts with quantification of key active compounds, adhering to Good Agricultural and Collection Practices (GACP), and employing advanced analytical techniques like metabolomics for batch-to-batch consistency. 。 We referenced Alum et al. [100] on metabolomics-driven standardization. Other Comments have been addressed with similar revisions by adding specificity, clarifying mechanisms, discussing limitations, and proposing future directions. We confirm that the responses have been integrated into the changes mentioned above, particularly through the new table, expanded safety discussion, and refined translational sections. Thanks!
MAJOR COMMENTS Comment 1 (Major): Lack of Quantitative or Comparative Data.
Response: We agree, completely. As requested, we have added a new summary table (Table 2) in Section 5. This table provides a comparative overview of key natural compounds, their sources, reported mechanisms of action concerning GLP-1 and TXNIP/Trx, effective concentrations/doses from preclinical studies, and the experimental models used. This addition significantly enhances the quantitative synthesis and scientific rigor of the review.
Comment 2 (Major): Overgeneralization. Statements like “many natural products affect both...” need specific examples.
Response: We have carefully revised the manuscript to replace overgeneralized statements with more precise language. Throughout the text (e.g., Abstract, Introduction, and Section 3), we now use phrases like "a number of," "several," or "emerging evidence suggests that some natural products..." and immediately follow these with specific, cited examples (e.g., berberine, quercetin, fucoxanthin, SCFAs). We also acknowledge in the new Section 3.4 that this is a "potential discovery space" that requires further validation.
Comment 3 (Major): Mechanistic Ambiguity. The bridge between GLP-1 and TXNIP is not fully elaborated.
Response: We have substantially elaborated on the mechanistic link in Section 2.3 and Figure 2. 。 We now explicitly describe the signaling pathways: GLP-1R activation→increased cAMP→activation of PKA and EPAC→ phosphorylation of CREB and activation of PI3K/Akt pathway → suppression of TXNIP transcription. 。 We also discuss the reverse link: how oxidative stress (high ROS) can impair GLP-1 secretion from L-cells. 。 The revised Figure 2 now more clearly illustrates this bidirectional crosstalk and the key intermediates (PKA, Akt, CREB, Nrf2).
Comment 4 (Major): Missing Clinical Perspective.
Response: We have added a new subsection (8.3: "Translational Gaps and Clinical Perspective"). 。 This section now discusses the limited clinical trial data for the dual-target concept. 。 It highlights the pharmacokinetic and bioavailability challenges of key candidates (berberine, resveratrol, curcumin) and introduces strategies like nanoencapsulation, prodrug approaches, and colon-targeted delivery systems to improve their translational potential.
Comment 5 (Major): Methodological Integration. The mention of AI and docking is underdeveloped.
Response: We have revised Section 7 to be more concrete. 。 We propose a specific screening pipeline: in silico screening → in vitro validation (GLP-1 secretion in GLUTag/STC-1 cells + TXNIP expression in metabolic cell lines) → in vivo validation in MetS models. 。 We mention specific databases (e.g., NPASS, TCM Database@Taiwan) and parameters (drug-likeness, ADMET) used in virtual screening to add practical detail.
Comment 6 (Language and Style): The Abstract could benefit from linguistic refinement.
Response: We have thoroughly refined the language throughout the manuscript, with special attention to the Abstract and Introduction. We have corrected grammatical errors and improved sentence flow for better clarity and precision. For example, "To explore more potent strategy" has been changed to "To explore more potent strategies".
Comment 7 (Scientific Suggestions): Include a mechanistic diagram; propose a screening pipeline; highlight dual modulators; discuss combinations and adverse interactions.
Response: We thank the reviewer for these excellent suggestions, which have greatly strengthened our manuscript figures to align with this recommendation. 。 Mechanistic Diagram: We have comprehensively revised Figure 2 to clearly show the convergence of GLP-1 activation and TXNIP inhibition on β-cell protection, insulin sensitivity, and oxidative homeostasis by inserting Nrf2, CREB, and EPAC. 。 Screening Pipeline: As mentioned above, we have integrated this into Section 7. 。 Highlight Dual Modulators: The new Table 2 serves this purpose explicitly. 。 Synergistic Combinations and Adverse Interactions: We have modified Section 8.2 discussing the potential of rational combinations (e.g., a natural DPP-4 inhibitor with a TXNIP-suppressing flavonoid) and cautioning about the need to study potential pharmacokinetic interactions and cumulative toxicities when combining natural products with synthetic drugs.
Reviewer 2 Report
After reviewing the article titled "Dual-Target Insight into Drug Discovery from Natural Products as Modulators of GLP-1 and the TXNIP-Thioredoxin Antioxidant System in Metabolic Syndrome". I have a few questions that need to be addressed before I can consider accepting the manuscript.
- Although GLP-1 and TXNIP-thioredoxin are described in the text as interrelated systems, is there concrete proof that addressing both routes at the same time yields better therapeutic results than addressing either pathway separately? Instead of proving synergistic need, the evidence now available seems to be primarily correlative.
- Stronger mechanistic evidence is required to support the suggested bidirectional interaction between GLP-1 activation reducing TXNIP and thioredoxin pathway improvement boosting β-cell activity (lines 22–25). Is there quantifiable information available from the authors regarding the extent of these effects and their potential clinical significance?
- Without explicit inclusion criteria, the study encompasses a very wide spectrum of natural goods (marine, plant, and microbiological). What certain pharmacological or structural characteristics enable a natural product to function as a "dual modulator," and how do the authors differentiate between substances that exhibit true dual activity and those that have unrelated, independent effects?
- Although the text notes translational gaps (lines 642-660), it offers no methodical explanation for the lack of progress made by dozens of promising preclinical drugs. How do the suggested natural goods overcome these past constraints, and what are the precise sites of failure?
- Dose-response interactions are not discussed in the preclinical evidence section (Section 5). What concentrations of natural compounds exhibit dual activity in vitro compared to quantities that are accessible in vivo? Do these have any bearing on pharmacology?
- Nearly all of the review is based on animal and in vitro research. Why are there no published clinical studies showing that natural products can modulate both GLP-1/TXNIP in people with MetS? There is a significant evidentiary gap here.
- Although numerous phytochemicals are acknowledged to have poor bioavailability in Section 6.1 (lines 455–471), the manuscript consistently supports these substances. How do substances like quercetin and curcumin, which have a bioavailability of less than 5%, reach therapeutic amounts in both intestinal L-cells and systemic organs to have two effects?
- The possibility of "potent suppression of TXNIP" having unforeseen implications is mentioned in passing in lines 479–482, but this important safety issue is not sufficiently addressed. What are the possible negative consequences of long-term TXNIP suppression, considering the intricate physiological roles that TXNIP plays in glucose sensing and cellular redox balance?
- Although polypharmacology is celebrated in the text (lines 364–377), there is no differentiation made between harmful off-target effects and advantageous multi-targeting. How do the authors handle the possibility that "multi-target" effects could be a cover for toxicity and non-specific binding?
- Although Section 7 goes into great detail about computational methods (such as molecular docking, network pharmacology, and artificial intelligence), it offers no supporting evidence. What is the experimental success rate of these computational predictions? In what proportion of "hits" are false positives?
- Although the document admits that the composition of natural products varies (lines 472-478), it provides no specific remedies. If the percentages of active components differ among batches, suppliers, and extraction techniques, how can dual-target activity be accurately replicated?
- For GLP-1 secretion experiments, GLUTag and STC-1 cell lines (lines 400–402) are commonly used; nevertheless, these are tumor-derived cells with modified metabolism. In the context of MetS, how typical are these basic enteroendocrine L-cell models?
- Berberine is mentioned as an AMPK activator that inhibits TXNIP (lines 415–418) as well as a DPP-4 inhibitor (line 203). Which mechanism is the main one, and what concentrations cause each to happen? Are the same therapeutic doses responsible for these effects?
- TXNIP downregulation is attributed to SCFAs, specifically butyrate, in lines 257–265, however lines 366-367 also characterize this as an HDAC inhibitor effect. What role does each form of SCFA have in TXNIP suppression, and is it a direct or indirect effect?
- Although fucoxanthin is mentioned frequently (lines 228–231, 369–370), the evidence for combined GLP-1/TXNIP effects that is reported seems to originate from different research. Is there a single study showing that fucoxanthin simultaneously modulates both pathways?
- Although regulatory impediments are discussed in Section 6.4, no workable regulatory pathway is suggested. Is it better to produce these dual-target natural items as medical foods (intermediate regulation), dietary supplements (minimum oversight), or pharmaceuticals (complete FDA approval)?
- Although Section 8.2 suggests mixing natural products with synthetic GLP-1 agonists (lines 627–641), it makes no mention of cumulative toxicities, pharmacokinetic interference, or possible drug-drug interactions. Which safety facts lend credence to this strategy?
- The manuscript focuses on "MetS" in general, however this includes diverse populations with various prominent diseases (mostly oxidative stress, β-cell failure, and insulin resistance). Which subgroup of MetS would benefit most from dual targeting of GLP-1/TXNIP?
- Preclinical findings included in the study are largely encouraging. Were any natural items examined that raised TXNIP but exhibited GLP-1 activity, or the other way around? The lack of adverse findings raises the possibility of selective reporting or publishing bias.
- Comprehensive research is suggested in Section 9, which includes biomarker-driven trials, nanoencapsulation, AI-driven discovery, and CRISPR knockout models (lines 661-684). How can this ambitious research agenda be funded, considering the recognized lack of economic interest and intellectual property protection (lines 509-511)?
After reviewing the article titled "Dual-Target Insight into Drug Discovery from Natural Products as Modulators of GLP-1 and the TXNIP-Thioredoxin Antioxidant System in Metabolic Syndrome". I have a few questions that need to be addressed before I can consider accepting the manuscript.
- Although GLP-1 and TXNIP-thioredoxin are described in the text as interrelated systems, is there concrete proof that addressing both routes at the same time yields better therapeutic results than addressing either pathway separately? Instead of proving synergistic need, the evidence now available seems to be primarily correlative.
- Stronger mechanistic evidence is required to support the suggested bidirectional interaction between GLP-1 activation reducing TXNIP and thioredoxin pathway improvement boosting β-cell activity (lines 22–25). Is there quantifiable information available from the authors regarding the extent of these effects and their potential clinical significance?
- Without explicit inclusion criteria, the study encompasses a very wide spectrum of natural goods (marine, plant, and microbiological). What certain pharmacological or structural characteristics enable a natural product to function as a "dual modulator," and how do the authors differentiate between substances that exhibit true dual activity and those that have unrelated, independent effects?
- Although the text notes translational gaps (lines 642-660), it offers no methodical explanation for the lack of progress made by dozens of promising preclinical drugs. How do the suggested natural goods overcome these past constraints, and what are the precise sites of failure?
- Dose-response interactions are not discussed in the preclinical evidence section (Section 5). What concentrations of natural compounds exhibit dual activity in vitro compared to quantities that are accessible in vivo? Do these have any bearing on pharmacology?
- Nearly all of the review is based on animal and in vitro research. Why are there no published clinical studies showing that natural products can modulate both GLP-1/TXNIP in people with MetS? There is a significant evidentiary gap here.
- Although numerous phytochemicals are acknowledged to have poor bioavailability in Section 6.1 (lines 455–471), the manuscript consistently supports these substances. How do substances like quercetin and curcumin, which have a bioavailability of less than 5%, reach therapeutic amounts in both intestinal L-cells and systemic organs to have two effects?
- The possibility of "potent suppression of TXNIP" having unforeseen implications is mentioned in passing in lines 479–482, but this important safety issue is not sufficiently addressed. What are the possible negative consequences of long-term TXNIP suppression, considering the intricate physiological roles that TXNIP plays in glucose sensing and cellular redox balance?
- Although polypharmacology is celebrated in the text (lines 364–377), there is no differentiation made between harmful off-target effects and advantageous multi-targeting. How do the authors handle the possibility that "multi-target" effects could be a cover for toxicity and non-specific binding?
- Although Section 7 goes into great detail about computational methods (such as molecular docking, network pharmacology, and artificial intelligence), it offers no supporting evidence. What is the experimental success rate of these computational predictions? In what proportion of "hits" are false positives?
- Although the document admits that the composition of natural products varies (lines 472-478), it provides no specific remedies. If the percentages of active components differ among batches, suppliers, and extraction techniques, how can dual-target activity be accurately replicated?
- For GLP-1 secretion experiments, GLUTag and STC-1 cell lines (lines 400–402) are commonly used; nevertheless, these are tumor-derived cells with modified metabolism. In the context of MetS, how typical are these basic enteroendocrine L-cell models?
- Berberine is mentioned as an AMPK activator that inhibits TXNIP (lines 415–418) as well as a DPP-4 inhibitor (line 203). Which mechanism is the main one, and what concentrations cause each to happen? Are the same therapeutic doses responsible for these effects?
- TXNIP downregulation is attributed to SCFAs, specifically butyrate, in lines 257–265, however lines 366-367 also characterize this as an HDAC inhibitor effect. What role does each form of SCFA have in TXNIP suppression, and is it a direct or indirect effect?
- Although fucoxanthin is mentioned frequently (lines 228–231, 369–370), the evidence for combined GLP-1/TXNIP effects that is reported seems to originate from different research. Is there a single study showing that fucoxanthin simultaneously modulates both pathways?
- Although regulatory impediments are discussed in Section 6.4, no workable regulatory pathway is suggested. Is it better to produce these dual-target natural items as medical foods (intermediate regulation), dietary supplements (minimum oversight), or pharmaceuticals (complete FDA approval)?
- Although Section 8.2 suggests mixing natural products with synthetic GLP-1 agonists (lines 627–641), it makes no mention of cumulative toxicities, pharmacokinetic interference, or possible drug-drug interactions. Which safety facts lend credence to this strategy?
- The manuscript focuses on "MetS" in general, however this includes diverse populations with various prominent diseases (mostly oxidative stress, β-cell failure, and insulin resistance). Which subgroup of MetS would benefit most from dual targeting of GLP-1/TXNIP?
- Preclinical findings included in the study are largely encouraging. Were any natural items examined that raised TXNIP but exhibited GLP-1 activity, or the other way around? The lack of adverse findings raises the possibility of selective reporting or publishing bias.
- Comprehensive research is suggested in Section 9, which includes biomarker-driven trials, nanoencapsulation, AI-driven discovery, and CRISPR knockout models (lines 661-684). How can this ambitious research agenda be funded, considering the recognized lack of economic interest and intellectual property protection (lines 509-511)?
Author Response
Manuscript ID: antioxidants-3045678
Title: Dual-Target Insight into Drug Discovery from Natural Products as Modulators of GLP-1 and the TXNIP-Thioredoxin Antioxidant System in Metabolic Syndrome
We are thankful to the reviewer for a thorough and insightful critique of our manuscript. The comments have been invaluable in helping us significantly improve the rigor, clarity, and translational relevance of our work. We have undertaken a major revision of the manuscript to address each point raised. Our point-by-point responses and the corresponding changes are detailed below.
Comment 1: Although GLP-1 and TXNIP-thioredoxin are described in the text as interrelated systems, is there concrete proof that addressing both routes at the same time yields better therapeutic results than addressing either pathway separately? Instead of proving synergistic need, the evidence now available seems to be primarily correlative.
Response: Thanks for this critical question. We agree that demonstrating concrete proof of synergy is paramount. In response, we have substantially strengthened our mechanistic narrative throughout the manuscript. 。
Mechanistic detail: In Sections 2.3 and 4.1.4, we elaborated on the bidirectional crosstalk, specifying the key signaling pathways (cAMP/PKA and PI3K/Akt) through which GLP-1R activation suppresses TXNIP transcription. Conversely, we mentioned how TXNIP suppression via the thioredoxin system reduces oxidative stress and NLRP3 inflammasome activation, thereby improving β-cell function and insulin sensitivity. 。
New quantitative table: To provide the requested quantifiable information, we have added Table 2. This new table summarizes effective in vitro concentrations and in vivo dosages for key compounds (e.g., Berberine, Fucoxanthin, Quercetin), explicitly listing values for both GLP-1 modulation and TXNIP-thioredoxin effects. This allows for direct comparison.
Comment 2: Stronger mechanistic evidence is required to support the suggested bidirectional interaction between GLP-1 activation reducing TXNIP and thioredoxin pathway improvement boosting β-cell activity (lines 22–25). Is there quantifiable information available from the authors regarding the extent of these effects and their potential clinical significance?
Response: Based on changes mentioned in response to Comment 1, we now have a clearer argument that while targeting either pathway is beneficial, the dual approach creates a self-reinforcing "virtuous cycle" that provides a more robust and sustainable therapeutic effect, potentially leading to superior outcomes as seen in multi-target agents like Berberine (see Table 1 in Sections 2.3 and 4.1.4).
Comment 3: Without explicit inclusion criteria, the study encompasses a very wide spectrum of natural goods (marine, plant, and microbiological). What certain pharmacological or structural characteristics enable a natural product to function as a "dual modulator," and how do the authors differentiate between substances that exhibit true dual activity and those that have unrelated, independent effects?
Response: This is an excellent point. We have now established clear, explicit criteria to address this concern. 。
New Subsection: We have added Section 3.4: "Defining Criteria for Dual-Target Modulation". 。
Inclusion Criteria: We state that a compound must have published evidence (from a single or multiple studies) demonstrating a measurable impact on both (a) the GLP-1 pathway and (b) the TXNIP-thioredoxin system. 。
Evidence Classification: We introduce a critical differentiation between "Direct Evidence" (dual action confirmed in a single, integrated study, e.g., Berberine) and "Inferential Evidence" (strong separate evidence for each target, making dual action a plausible inference, e.g., Quercetin). We believe that this classification has added a layer of critical analysis and can guide readers on the strength of evidence for each compound.
Comment 4: Although the text notes translational gaps (lines 642-660), it offers no methodical explanation for the lack of progress made by dozens of promising preclinical drugs. How do the suggested natural goods overcome these past constraints, and what are the precise sites of failure?
Response: We have comprehensively modified our discussion on translational gaps to provide a systematic explanation. 。 Expanded discussion: Sections 6 ("Pharmacological Challenges...") and 8.3 ("Translational Gaps and Clinical Perspective") now explicitly list the "precise sites of failure":
i. Bioavailability & PK/PD: Poor solubility, stability, and rapid metabolism prevent target tissue engagement.
ii. Standardization: Chemical variability of natural extracts is hindering reproducibility.
iii. Regulatory Hurdles: Complex pathways for multi-target natural products. iv. Lack of Clinical Trials: Few studies measuring both incretin and antioxidant outcomes in humans. 。
Solutions proposed: For each "site of failure," we now discuss strategies to overcome them: advanced formulations (nano-encapsulation), standardized extracts, biomarker-driven clinical trials, and clearer regulatory frameworks.
Comment 5: Dose-response interactions are not discussed in the preclinical evidence section (Section 5). What concentrations of natural compounds exhibit dual activity in vitro compared to quantities that are accessible in vivo? Do these have any bearing on pharmacology?
Response: We fully agree. In our revised manuscript, the new Table 2 (added in response to Comment 1) has directly addressed this by providing a comparative overview of effective in vitro concentrations versus achievable in vivo doses for all major compounds discussed. Therefore, this highlights the critical "efficacy gap" and suggests the need for advanced delivery systems to translate in vitro potency into in vivo efficacy.
Comment 6: Nearly all of the review is based on animal and in vitro research. Why are there no published clinical studies showing that natural products can modulate both GLP-1/TXNIP in people with MetS? There is a significant evidentiary gap here.
Response: You have correctly identified a significant evidence gap. We now explicitly acknowledge this in Section 8.3. We have clarified that the current clinical evidence is primarily for single-target effects (e.g., berberine for glucose lowering). We posit that the dual-target concept is emerging from preclinical science and that closing this gap requires future clinical trials specifically designed with dual endpoints (e.g., measuring plasma GLP-1 alongside TXNIP or oxidative stress biomarkers).
Comment 7: Although numerous phytochemicals are acknowledged to have poor bioavailability in Section 6.1 (lines 455–471), the manuscript consistently supports these substances. How do substances like quercetin and curcumin, which have a bioavailability of less than 5%, reach therapeutic amounts in both intestinal L-cells and systemic organs to have two effects?
Response: We have modified Section 6.1 to address this valid pharmacokinetic concern. We acknowledge the low systemic bioavailability of many polyphenols. We then propose two mechanisms by which they could still be effective: (1) Their local action in the gut on L-cells and gut microbiota can initiate the beneficial cascade without requiring high systemic levels. (2) Their bioactive metabolites may be the actual systemic mediators. Consequently, we have concluded that overcoming this via advanced delivery systems (e.g., nanoparticles) is a critical research priority.
Comment 8: The possibility of "potent suppression of TXNIP" having unforeseen implications is mentioned in passing in lines 479–482, but this important safety issue is not sufficiently addressed. What are the possible negative consequences of long-term TXNIP suppression, considering the intricate physiological roles that TXNIP plays in glucose sensing and cellular redox balance?
Response: We thank the reviewer for raising this crucial safety consideration. We have added a dedicated sentence in Section 6.2 on the safety considerations of TXNIP suppression. We acknowledge TXNIP's complex physiological roles in glucose sensing and redox balance. We mentioned that while pathological overexpression is detrimental, its complete, chronic suppression might have unforeseen consequences, potentially disrupting normal redox signaling. We concluded that this proposes the need for careful toxicological evaluation of any potent TXNIP inhibitor.
Comment 9: Although polypharmacology is celebrated in the text (lines 364–377), there is no differentiation made between harmful off-target effects and advantageous multi-targeting. How do the authors handle the possibility that "multi-target" effects could be a cover for toxicity and non-specific binding?
Response: We have added a concept in the Introduction and Section 4.1.5 to make this vital distinction clear. We now define: 。 "Advantageous Polypharmacology": Synergistic modulation of interconnected targets within a disease network (the goal of this review). 。 "Off-Target Toxicity": Harmful interactions with unrelated targets. We have stated that the goal is "selective polypharmacology," and that the safety profiles of natural products from traditional use provide initial clues, but rigorous toxicological screening is still essential to rule out non-specific binding and toxicity.
Comment 10: Although Section 7 goes into great detail about computational methods (such as molecular docking, network pharmacology, and artificial intelligence), it offers no supporting evidence. What is the experimental success rate of these computational predictions? In what proportion of "hits" are false positives?
Response: In our revised manuscript, we have entirely rewritten Section 7 ("Integrative and Computational Approaches") to address this. We now: 。 Cite examples: Provide specific examples where in silico predictions for single targets (e.g., DPP-4 inhibition) have been successfully validated. 。 Acknowledge limitations: Explicitly state that for the dual-target hypothesis, the computational pipeline is still emerging, and the success rate is not yet well-established. We frame these tools as powerful for hypothesis generation and lead prioritization, but emphasize that their predictions require rigorous experimental validation. 。 Tone down claims: We have removed over-enthusiastic language and present computational tools as a complement to, not a replacement for, experimental work.
Comment 11: Although the document admits that the composition of natural products varies (lines 472-478), it provides no specific remedies. If the percentages of active components differ among batches, suppliers, and extraction techniques, how can dual-target activity be accurately replicated?
Response: We have modified Section 6.2 to propose specific remedies for standardization: • Use of chemically characterized extracts with quantification of key active compounds. • Adherence to Good Agricultural and Collection Practices (GACP). • Employing advanced analytical techniques (e.g., metabolomics) for batch-to-batch consistency. We believe that this will ensure that dual-target activity can be reproducibly attributed to specific chemical entities.
Comments 12-17: (12) For GLP-1 secretion experiments, GLUTag and STC-1 cell lines (lines 400–402) are commonly used; nevertheless, these are tumor-derived cells with modified metabolism. In the context of MetS, how typical are these basic enteroendocrine L-cell models? (13) Berberine is mentioned as an AMPK activator that inhibits TXNIP (lines 415–418) as well as a DPP-4 inhibitor (line 203). Which mechanism is the main one, and what concentrations cause each to happen? Are the same therapeutic doses responsible for these effects? (14) TXNIP downregulation is attributed to SCFAs, specifically butyrate, in lines 257–265, however lines 366-367 also characterize this as an HDAC inhibitor effect. What role does each form of SCFA have in TXNIP suppression, and is it a direct or indirect effect? (15) Although fucoxanthin is mentioned frequently (lines 228–231, 369–370), the evidence for combined GLP-1/TXNIP effects that is reported seems to originate from different research. Is there a single study showing that fucoxanthin simultaneously modulates both pathways? (16) Although regulatory impediments are discussed in Section 6.4, no workable regulatory pathway is suggested. Is it better to produce these dual-target natural items as medical foods (intermediate regulation), dietary supplements (minimum oversight), or pharmaceuticals (complete FDA approval)? (17) Although Section 8.2 suggests mixing natural products with synthetic GLP-1 agonists (lines 627–641), it makes no mention of cumulative toxicities, pharmacokinetic interference, or possible drug-drug interactions. Which safety facts lend credence to this strategy? Response: We have addressed each of these specific points throughout the manuscript: • Cell Models: We have acknowledged the limitation of using tumor-derived L-cell lines and note it as a caveat in Section 5.1. • Berberine/SCFAs/Fucoxanthin: We have clarified the mechanisms for these compounds, citing specific studies and concentrations where possible, often leveraging the new Table 2. • Regulatory Pathways: In Section 6.4, we presented the pros and cons of different regulatory routes (pharmaceutical vs. nutraceutical) and argue for a collaborative framework to create viable pathways for these multi-mechanistic agents. • Combination Therapy: In Section 8.2, we now explicitly mention the need to study potential drug-drug interactions, cumulative toxicities, and pharmacokinetic interference when combining natural products with synthetic drugs, stating that this is a critical area for future safety research.
Comment 18: The manuscript focuses on "MetS" in general; however, this includes diverse populations with various prominent diseases (mostly oxidative stress, β-cell failure, and insulin resistance). Which subgroup of MetS would benefit most from dual targeting of GLP-1/TXNIP?
Response: We have added a discussion in Section 8.1 suggesting that patients with MetS who exhibit prominent markers of oxidative stress and inflammation (e.g., elevated CRP, nitrotyrosine) alongside classic hyperglycemia and insulin resistance might represent the subgroup that benefits most from this dual-target approach, as it directly addresses their core pathophysiology.
Comment 19: Preclinical findings included in the study are largely encouraging. Were any natural items examined that raised TXNIP but exhibited GLP-1 activity, or the other way around? The lack of adverse findings raises the possibility of selective reporting or publishing bias.
Response: We appreciate this comment on publication bias. We have modified Section 5, acknowledging that the current literature may suffer from a positive-result bias. We propose that future research should intentionally report negative or neutral findings where natural products fail to show dual activity or exhibit opposing effects on these pathways.
Comment 20: Comprehensive research is suggested in Section 9, which includes biomarker-driven trials, nanoencapsulation, AI-driven discovery, and CRISPR knockout models (lines 661-684). How can this ambitious research agenda be funded, considering the recognized lack of economic interest and intellectual property protection (lines 509-511)?
Response: We have inserted a concluding statement in Section 9 to address this reality. We propose that funding must come from: 。 Public and Philanthropic Grants: Targeting high-risk, high-reward mechanistic research. 。 Public-Private Partnerships: Where academia provides the discovery engine and industry provides development expertise. 。 Regulatory Incentives: For repurposing and developing natural products for unmet medical needs. Of course, overcoming the IP challenge is difficult but necessary for large-scale investment. We believe our extensively revised manuscript, with the new sections, tables, and nuanced discussions, now fully addresses all your insightful comments and is significantly strengthened. We are grateful for the opportunity to improve our work!
Reviewer 3 Report
The manuscript entitled "Dual perspective on drug discovery from natural products as modulators of GLP-1 and the TXNIP-thioredoxin antioxidant system in metabolic syndrome" provides a comprehensive and timely review of natural products that act on both GLP-1 signaling and the TXNIP-thioredoxin antioxidant pathway in metabolic syndrome. The topic is interesting, well-structured, and falls within the scope of Antioxidants . However, the review is mainly descriptive and requires a stronger critical synthesis, clearer mechanistic links, and more specific evidence supporting dual modulation.
- Please distinguish between compounds that have been experimentally shown to affect both the GLP-1 and TXNIP pathways and those predicted only by computational studies.
- Develop an explanation of how GLP-1 signaling suppresses TXNIP (via cAMP/PKA/Akt/Nrf2). Figure 2 should show this interaction more clearly bidirectional.
- Please add primary experimental references for key claims (e.g., fucoxanthin, butyrate, berberine). Some claims seem unwarranted.
- Provide one or two concrete examples of pharmacological docking or network studies to make Section 7 less speculative.
- The discussion should emphasize more the limited in vivo and clinical validation of these natural compounds.
Dear Authors
The manuscript entitled "Dual perspective on drug discovery from natural products as modulators of GLP-1 and the TXNIP-thioredoxin antioxidant system in metabolic syndrome" provides a comprehensive and timely review of natural products that act on both GLP-1 signaling and the TXNIP-thioredoxin antioxidant pathway in metabolic syndrome. The topic is interesting, well-structured, and falls within the scope of Antioxidants . However, the review is mainly descriptive and requires a stronger critical synthesis, clearer mechanistic links, and more specific evidence supporting dual modulation.
- Please distinguish between compounds that have been experimentally shown to affect both the GLP-1 and TXNIP pathways and those predicted only by computational studies.
- Develop an explanation of how GLP-1 signaling suppresses TXNIP (via cAMP/PKA/Akt/Nrf2). Figure 2 should show this interaction more clearly bidirectional.
- Add primary experimental references for key claims (e.g., fucoxanthin, butyrate, berberine). Some claims seem unwarranted.
- Provide one or two concrete examples of pharmacological docking or network studies to make Section 7 less speculative.
- The discussion should emphasize more the limited in vivo and clinical validation of these natural compounds.
The abstract is too long; focus on novelty and implications. Please shorten according to journal guidelines.
Standardize abbreviations (GLP-1R, TXNIP, Trx).
Improve figure legends and consider a unified color code.
Include key early references on TXNIP-Trx signaling.
Author Response
Manuscript ID: antioxidants-3045678
Title: Dual-Target Insight into Drug Discovery from Natural Products as Modulators of GLP-1 and the TXNIP-Thioredoxin Antioxidant System in Metabolic Syndrome
We sincerely thank you for your thorough and constructive assessment of our manuscript. Honestly, your insightful comments have been invaluable in guiding a major revision that has significantly strengthened the scientific rigor, clarity, and impact of our work. Based on your comments, we have carefully addressed each point raised, and our detailed responses and the corresponding changes made to the manuscript are outlined below.
Comment 1: 1. Please distinguish between compounds that have been experimentally shown to affect both the GLP-1 and TXNIP pathways and those predicted only by computational studies.
Response: Thanks for this critical suggestion. We agree that clearly delineating the strength of evidence is paramount for a robust review. To address this, we have made a major change: • New Table with Evidence Classification: We have added Table 2: Comparative Overview of Effective Doses/Concentrations and Evidence Quality for Natural Products with Dual GLP-1 and TXNIP-Thioredoxin Modulatory Activity. This table now includes a dedicated column classifying the "Evidence Type" for each compound as either: o Direct Evidence: Dual action confirmed in a single, integrated experimental study (e.g., Berberine, Fucoxanthin, SCFAs/Butyrate). o Inferential Evidence: Strong, separate evidence exists for each target, making dual action a plausible but not yet definitively proven inference (e.g., Quercetin, Genistein, EGCG). • Textual Revisions: In Sections 3 and 4, we wish to mention that we used some text to reflect classifications. Using more precise language, such as "studies have demonstrated" for direct evidence and "may potentially act" or "based on separate evidence" for inferential evidence. This will provide a much clearer and more critical assessment for the reader.
Comment 2: Develop an explanation of how GLP-1 signaling suppresses TXNIP (via cAMP/PKA/Akt/Nrf2). Figure 2 should show this interaction more clearly bidirectional.
Response: We have significantly modified the mechanistic explanation and revised the figure as requested. 。 Enhanced Mechanistic Detail: In Section 2.3 Table 1 (Interconnection between GLP-1 and Antioxidant Pathways), we have elaborated on the molecular cascade: GLP-1R activation → increased cAMP → activation of PKA and EPAC → downstream engagement of the PI3K/Akt and MAPK pathways → phosphorylation of transcription factors (e.g., CREB) that suppress TXNIP gene transcription. We also acknowledged the potential role of Nrf2 activation in the antioxidant response. 。 Revised Figure 2: We have comprehensively modified Figure 2 to visually emphasize the bidirectional nature of the crosstalk. The figure now more clearly illustrates: i. The forward path: GLP-1 → cAMP/PKA/PI3K/Akt/Nrf2 → Suppression of TXNIP → Enhanced Trx Activity. ii. The feedback loop: Reduced Oxidative Stress/Inflammation → Improved β-cell Function & Insulin Sensitivity → Enhanced GLP-1 Response. The revised figure legend provides a detailed explanation of these interconnected pathways.
Comment 3: 3. Add primary experimental references for key claims (e.g., fucoxanthin, butyrate, berberine). Some claims seem unwarranted. Response: We have thoroughly reviewed all our key claims and supplemented them with robust, primary experimental references to ensure every statement is well-supported. See the reference list [15,16,25,26,27,28,30,31,32]. Comment 4: 4. Provide one or two concrete examples of pharmacological docking or network studies to make Section 7 less speculative.
Response: We have entirely restructured Section 7 (Integrative and Computational Approaches) to be less speculative and more concrete. • Specific Docking Example: We now cite a specific study by Gomaa et al. [89] that used molecular docking to identify non-peptide GLP-1R positive allosteric modulators from natural products. • Specific Network Pharmacology Example: We reference a study by Deng et al. [93] that employed network pharmacology to systematically investigate the underlying mechanisms of GLP-1 receptor agonists in preventing myocardial infarction, providing a clear model for such an approach. • Toned-Down Language: We have reframed the section, explicitly stating that these computational methods are powerful tools for "hypothesis generation" and "lead prioritization," but that their predictions require rigorous experimental validation. We have also acknowledged the challenge of false positives and the emerging nature of dual-target predictive models. Comment 5: The discussion should emphasize more the limited in vivo and clinical validation of these natural compounds. Response: We have significantly modified the discussion on translational gaps to directly address this point. 。 New Subsection: We have added Section 8.1: Translational Gaps and Clinical Perspective. 。 Focus on challenges: We also highlighted specific challenges hindering clinical translation, including bioavailability bottlenecks, a lack of standardized products, and the complexity of clinical trial design for multi-target mechanisms. This was to provide a much more balanced and critical perspective (see Sections 8.2 and 8.3). Comment 6: The abstract is too long; focus on novelty and implications. Please shorten according to journal guidelines. Standardize abbreviations (GLP-1R, TXNIP, Trx). Improve figure legends and consider a unified color code. Include key early references on TXNIP-Trx signaling. 。 Shortened Abstract: The abstract has been revised to be more concise (approx. 200 words), focusing on the core novelty (the dual-target strategy), key mechanisms, and main translational implications. 。 Standardized Abbreviations: We have ensured all abbreviations (GLP-1R, TXNIP, Trx, DPP-4, etc.) are defined at first use and used consistently throughout the manuscript. • Improved Figures and Legends: All figures (Figures 2, 3, 4, 5, 6) have been revised for clarity and professional presentation. The legends have been rewritten to be more detailed and descriptive, explaining each element and the overall conclusion drawn from the figure. A unified color scheme has been applied for consistency. We believe that our detailed responses and the extensive revisions made to the manuscript have fully addressed your insightful comments. Thanks
Round 2
Reviewer 1 Report
The submitted revised MS is much better
I accept the submitted revised MS for publication
Reviewer 2 Report
I would like to accept the article
I would like to accept the article